# GOOD: Exploring Geometric Cues for Detecting Objects in an Open World

**Haiwen Huang**[1,2]   **Andreas Geiger**[2,3]   **Dan Zhang**[1,4]

[1]Bosch Lab, University of Tübingen
[2]Autonomous Vision Group, University of Tübingen
[3]Tübingen AI Center  [4]Bosch Center for Artificial Intelligence
{haiwen.huang, a.geiger}@uni-tuebingen.de, Dan.Zhang2@de.bosch.com

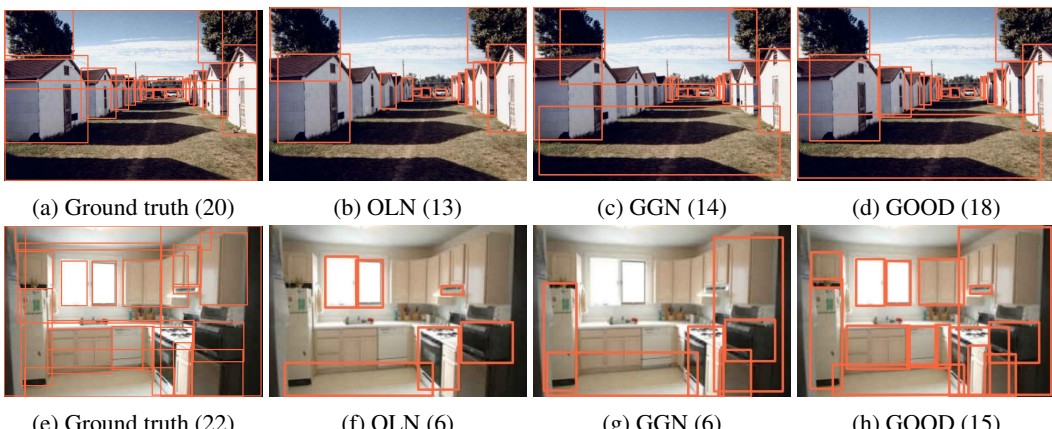

| (a) Ground truth (20) | (b) OLN (13) | (c) GGN (14) | (d) GOOD (18) |
| (e) Ground truth (22) | (f) OLN (6) | (g) GGN (6) | (h) GOOD (15) |

Figure 1: **Comparison of GOOD with different baselines.** Images in the first column are from validation sets of ADE20K (Zhou et al., 2019). From the second to fourth columns we show the detection results of three open-world object detection methods: OLN Kim et al. (2021), GGN Wang et al. (2022), and our Geometry-guided Open-world Object Detector (GOOD). The shown detection results are true-positive proposals from the top 100 proposals of each method. The numbers of true positive proposals or ground truth objects are denoted in parentheses. All models are trained on the RGB images from the PASCAL-VOC classes of the COCO dataset (Lin et al., 2014), which do not include houses, trees, or kitchen furniture. Both OLN and GGN fail to detect many objects not seen during training. GOOD generalizes better to unseen categories by exploiting the geometric cues.

## ABSTRACT

We address the task of open-world class-agnostic object detection, i.e., detecting every object in an image by learning from a limited number of base object classes. State-of-the-art RGB-based models suffer from overfitting the training classes and often fail at detecting novel-looking objects. This is because RGB-based models primarily rely on appearance similarity to detect novel objects and are also prone to overfitting short-cut cues such as textures and discriminative parts. To address these shortcomings of RGB-based object detectors, we propose incorporating geometric cues such as depth and normals, predicted by general-purpose monocular estimators. Specifically, we use the geometric cues to train an object proposal network for pseudo-labeling unannotated novel objects in the training set. Our resulting Geometry-guided Open-world Object Detector (GOOD) significantly improves detection recall for novel object categories and already performs well with only a few training classes. Using a single "person" class for training on the COCO dataset, GOOD surpasses SOTA methods by 5.0% AR@100, a relative improvement of 24%. The code has been made available at https://github.com/autonomousvision/good.

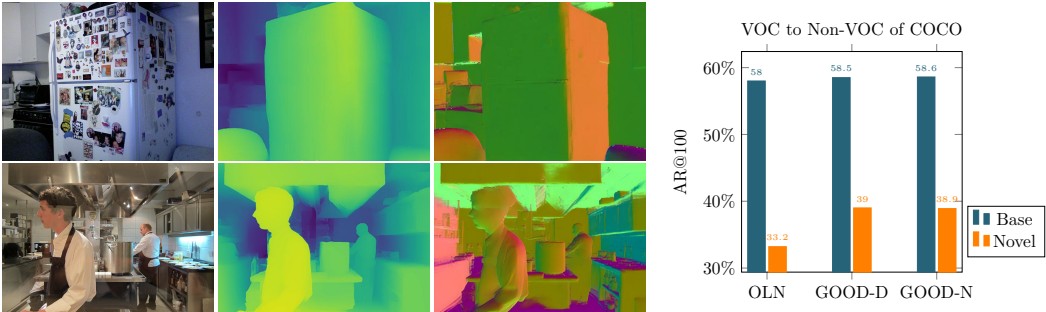

Figure 2: **Geometry cues are complementary to appearance cues for object localization.** The depth and normal cues of the RGB image are extracted using off-the-shelf general-purpose monocular predictors. **Left:** Geometric cues abstract away the appearance details and focus on more holistic information such as object shapes and relative spatial locations (depth) and directional changes (normals). **Right:** By incorporating geometric cues, GOODs generalize better than the RGB-based model OLN (Kim et al., 2021), i.e., much smaller AR gaps between the base and novel classes.

## 1 INTRODUCTION

The standard object detection task is to detect objects from a predefined class list. However, when deploying the model in the real world, it is rarely the case that the model will only encounter objects from its predefined taxonomy. In the open-world setup, object detectors are required to detect all the objects in the scene even though they have only been trained on objects from a limited number of classes. Current state-of-the-art object detectors typically struggle in the open-world setup. As a consequence, open-world object detection has gained increased attention over the last few years (Jaiswal et al., 2021; Kim et al., 2021; Joseph et al., 2021; Wang et al., 2022). In this work, we specifically address the task of open-world class-agnostic object detection, which is a fundamental task for downstream applications like open-world multi-object tracking (Liu et al., 2022), robotics (Jiang et al., 2019), and autonomous AI agents (Liu et al., 2021).

One reason for the failure of current object detectors in the open-world setting is that during training, they are penalized for detecting unlabeled objects in the background and are thus discouraged from detecting them. Motivated by this, previous works have designed different architectures (Kim et al., 2021; Konan et al., 2022) and training pipelines (Saito et al., 2021; Wang et al., 2022) to avoid suppressing the unannotated objects in the background, which have led to significant performance improvements. However, these methods still suffer from overfitting the training classes. Training only on RGB images, they mainly rely on appearance cues to detect objects of new categories and have great difficulty generalizing to novel-looking objects. Also, there are known short-cut learning problems with regard to training on RGB images Geirhos et al. (2019; 2020); Sauer & Geiger (2021) – there is no constraint for overfitting the textures or the discriminative parts of the known classes during training. In this work, we propose to tackle this challenge by incorporating geometry cues extracted by general-purpose monocular estimators from the RGB images. We show that such cues significantly improve detection recall for novel object categories on challenging benchmarks.

Estimating geometric cues such as depth and normals from a single RGB image has been an active research area for a long time. Such mid-level representations possess built-in invariance to many changes (e.g., brightness, color) and are more class-agnostic than RGB signals, see Figure 2. In other words, there is less discrepancy between known and unknown objects in terms of geometric cues. In recent years, thanks to stronger architectures and larger datasets (Ranftl et al., 2021b; 2022; Eftekhar et al., 2021), monocular estimators for mid-level representations have significantly advanced in terms of prediction quality and generalization to novel scenes. These models are able to compute high-quality geometric cues efficiently when used off-the-shelf as pre-trained models on new datasets. Therefore, it becomes natural to ask if these models can provide additional knowledge for current RGB-based open-world object detectors to overcome the generalization problem.

In this paper, we propose to use a pseudo-labeling method for incorporating geometric cues into open-world object detector training. We first train an object proposal network on the predicted

depth or normal maps to discover novel unannotated objects in the training set. The top-ranked novel object predictions are used as pseudo boxes for training the open-world object detector on the original RGB input. We observe that incorporating the geometry cues can significantly improve the detection recall of unseen objects, especially those that differ strongly from the training objects, as shown in Figure 1 and 2. We speculate that this is due to the complementary nature of geometry cues and the RGB-based detection cues: the geometry cues help discover novel-looking objects that RGB-based detectors cannot detect, and the RGB-based detectors can make use of more annotations with their strong representation learning ability to generalize to novel, unseen categories.

Our resulting Geometry-guided Open-world Object Detector (GOOD) surpasses the state-of-the-art performance on multiple benchmarks for open-world class-agnostic object detection. Thanks to the rich geometry information, GOOD can generalize to unseen categories with only a few known classes for training. Particularly, with a single training class "person" on the COCO dataset (Lin et al., 2014), GOOD can surpass SOTA methods by 5.0% AR@100 (a relative improvement of 24%) on detecting objects from non-person classes. With 20 PASCAL-VOC classes for training, GOOD surpasses SOTA methods even by 6.1% AR@100 in detecting non-VOC classes. Furthermore, we also analyze the advantages of geometric cues and show that they are less sensitive to semantic shifts across classes, and are better than other strategies for improving generalization.

## 2  RELATED WORK

**Open-world class-agnostic object detection** is the task of localizing all the objects in an image by learning with only a limited number of object classes (base classes). The core problem with standard object detection training is that the model is trained to classify the unannotated objects as background and thus is suppressed to detect them at inference time. To solve this issue, Kim et al. (2021) proposed object localization network (OLN), which replaces the classification heads of Faster RCNN (Ren et al., 2015) with class-agnostic objectness heads so that the training loss is only calculated on positive samples, i.e., known objects, and thus not suppressing the detection of unannotated novel objects. Saito et al. (2021) synthesized a training set by copy-pasting known objects onto synthetic backgrounds. However, the model struggles with the synthetic-to-real domain gap in solving the object detection task. Besides background non-suppression, a more proactive approach is to exploit unannotated objects for training. Wang et al. (2022) built upon traditional learning-free methods and developed a pairwise affinity predictor to discover unannotated objects. Their object detector, GGN, is then trained using the newly-discovered object masks and ground truth base class annotations as supervision. Finally, another promising direction is to use open-world knowledge from large pretrained multi-modal models. Recently, Minderer et al. (2022); Maaz et al. (2022) made use of pretrained language-vision model (Radford et al., 2021) to detect open-world objects using text queries. Our work is most related to OLN and GGN. We used the OLN architecture and training loss, but additionally incorporated geometry cues through our pseudo-labeling method. GGN used the pairwise affinity for pseudo labeling. However, since the pairwise affinity is trained on RGB inputs using the base class annotations, GGN still suffers from the overfitting problems of RGB-based methods. Our experiments showed geometric cues as a better source of pseudo boxes.

**Incorporating geometric cues for generalization.** The estimation of geometric cues has been an active research area for decades. With the introduction of deep neural networks, the seminal work by Eigen et al. (2014); Eigen & Fergus (2015) significantly improved over early works (Hoiem et al., 2005a;b; 2007; 2008; Saxena et al., 2005; 2008a;b). Recent progress in estimating geometric cues can be attributed to the use of modern architectures (Ranftl et al., 2022), stronger training strategies (Zamir et al., 2020) and large-scale datasets (Eftekhar et al., 2021). In particular, Omnidata (Eftekhar et al., 2021) has made significant headway in prediction quality and cross-dataset generalization. Since geometric cues abstract away the appearance details and retain more holistic information about the objects, such as shapes, they have been incorporated into many applications for generalization. For example, Xiang et al. (2022) incorporated them into the 3D shape completion pipeline to generalize to novel classes. Yu et al. (2022) used them to guide the optimization of neural implicit surface models for tackling scenes captured from sparse viewpoints. Chen et al. (2021) applied mid-level visual representations to reinforcement training and gained robustness under domain shifts. In this work, we propose to incorporate geometric cues through pseudo-labeling and also demonstrate large performance gains on open-world class-agnostic object detection benchmarks.

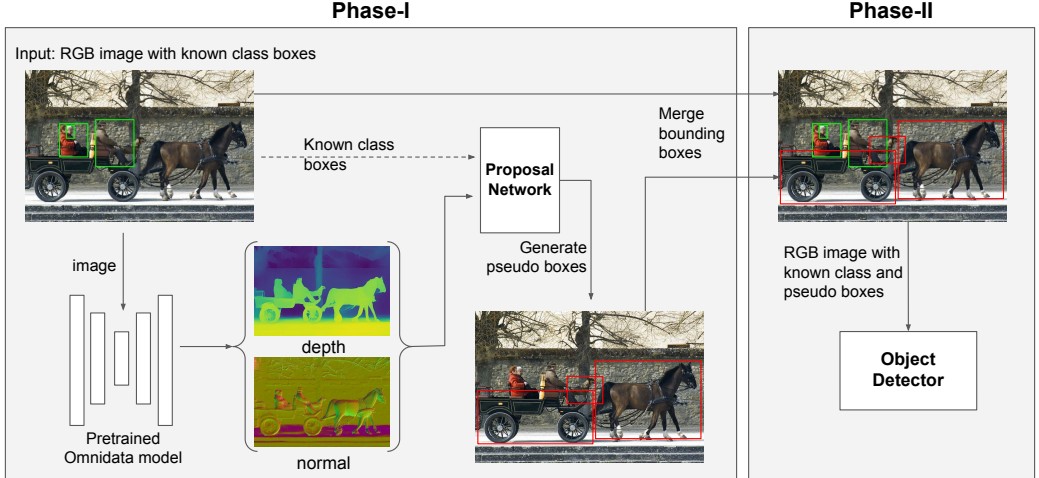

Figure 3: **Overview of the geometry-guided pseudo labeling method.** It consists of two training phases. Phase I: the RGB input is firstly preprocessed by the off-the-shelf model to extract the geometry cues, which are then used to train an object proposal network with the base class bounding box annotations. The proposal networks then pseudo-label the training samples, discovering unannotated novel objects. The top-ranked pseudo boxes are added to the annotation pool for Phase II training, i.e., a class-agnostic object detector is directly trained on the RGB input using both the base class and pseudo annotations. At inference time, we only need the model from Phase II.

# 3 METHOD

Our goal is to incorporate geometric cues for an improved open-world class-agnostic object detection performance. Concretely, we propose a pseudo labeling method, which can effectively utilize the geometric cues to detect unannotated novel objects in the training set and then use them for training the object detector. See the overview of our method in Figure 3.

## 3.1 OPEN-WORLD CLASS-AGNOSTIC OBJECT DETECTION PROBLEM

Current state-of-the-art object detection methods work well under the closed-world assumption. They are trained with a set of object bounding box annotations $\{t_i\}$ from a pre-specified list $\mathcal{K}$ of semantic classes, i.e., the base classes. At test time, their generalization is evaluated by detecting the objects from the known base classes. The standard training loss of object detection for an image $\mathcal{I}$ has two parts: classification loss and bounding box regression loss

$$\mathcal{L}(\mathcal{I}) = \frac{1}{N_{cls}} \sum_{i \in \mathcal{B}} \mathcal{L}_{cls}(p_i, p_i^*) + \frac{1}{N_{reg}} \sum_{i \in \mathcal{B}_{\mathcal{K}}} \mathcal{L}_{reg}(t_i, t_i^*), \tag{1}$$

where $i$ is the index of an anchor from the candidate set $\mathcal{B}$, $\{p_i, t_i\}$ are the predicted label and bounding box coordinates, and $\{p_i^*, t_i^*\}$ are the corresponding ground truths. $N_{cls}$ is the total number of anchors in the candidate set $\mathcal{B}$. $N_{reg}$ is the size of the subset $\mathcal{B}_{\mathcal{K}}$, which only contains the anchors with the ground truth label $p_i^* = 1$. Note $p_i^*$ equals 1 only when the anchor $i$ can be associated to an annotated object bounding box from the known classes $\mathcal{K}$; otherwise it equals 0 (background).

From the closed-world to the open-world setup, the generalization goal extends to localizing every object in the image, which can belong to an unknown novel class $u \in \mathcal{U}$. Under the training loss in (1), an unannotated object will be classified as "background". As a result, the model will treat similar types of objects as "background" at inference time. To avoid suppressing the detection of novel objects in the background, Kim et al. (2021) proposed to replace the classification loss in (1) by the objectness score prediction loss, yielding

$$\mathcal{L}_{OLN}(\mathcal{I}) = \frac{1}{N_{reg}} \sum_{i \in \mathcal{B}_{\mathcal{K}}} \mathcal{L}_{reg}(t_i, t_i^*) + \frac{1}{N_{reg}} \sum_{i \in \mathcal{B}_{\mathcal{K}}} \mathcal{L}_{obj}(o_i, o_i^*), \tag{2}$$

where $o_i$ and $o_i^*$ are the predicted objectness score and its ground truth of anchor $i$. In doing so, only the anchors with $p_i^* = 1$ are involved in training, completely removing any "background" prediction. At inference time, the objectness score is used to rank the detections. However, since these anchors only capture the annotated objects from the base classes, this loss modification cannot effectively mitigate the overfitting to the base classes. We further resort to adding additional "objects" into training, especially novel ones with very different appearances than objects from the base classes.

## 3.2 Exploiting geometric cues

Models trained on RGB images tend to over-rely on the appearance cues for object detection. Therefore, it is hard for them to detect novel objects that appear very differently from the base classes. For instance, a model trained on cars is likely to detect trucks, but unlikely to also detect sandwiches. Involving such novel objects, e.g., food, into training is then an effective way to mitigate the appearance bias towards the base classes, e.g., vehicles. To this end, we exploit two types of geometric cues, i.e., depth and normals, for detecting unannotated novel objects in the training set, see some examples in Figure 2. Both of them are common geometric cues that capture local information. Depth focuses on the relative spatial difference of objects and abstracts away the details on the object surfaces. Surface normals focus on the directional difference and remain the same on flat surfaces. Compared with the original RGB image, they discard most appearance details and focus on the geometry information such as object shapes and relative spatial locations. Models trained with them can thus discover many novel-looking objects that RGB-based ones cannot detect.

We use off-the-shelf pretrained models to extract geometric cues. Specifically, we use Omnidata models (Eftekhar et al., 2021) trained using cross-task consistency (Zamir et al., 2020) and 2D/3D data augmentations (Kar et al., 2022). The training dataset for the models is the Omnidata Starter Dataset (OSD) (Eftekhar et al., 2021) which contains 2200 real and rendered scenes. Despite the difference between the OSD to the object detection benchmark datasets, the Omnidata model can produce high-quality results, implying that the invariances behind these geometric cues are robust.

## 3.3 Pseudo labeling method

To use the geometric cues to discover unannotated novel objects in the training set, we first train an object proposal network on the depth or normal input using the same training loss as in (2), i.e., Phase-I training in Figure 3. Then, this object proposal network will pseudo-label the training images using its detected bounding boxes. After filtering out the detected bounding boxes which overlap with the base class annotations, we then add the remaining top-$k$ boxes to the ground truth annotations. Here $k \in \{1, 2, 3\}$ is determined for each detector on a small holdout validation set. Finally, we train a new class-agnostic object detector using the RGB image as input and the extended bounding box annotation pool as ground truth, i.e., Phase-II in Figure 3. The training loss is

$$\mathcal{L}_{GOOD}(\mathcal{I}) = \frac{1}{N_{reg}} \sum_{i \in \mathcal{B}_\mathcal{K} \cup \mathcal{B}_\mathcal{N}} \mathcal{L}_{reg}(t_i, t_i^*) + \frac{1}{N_{reg}} \sum_{i \in \mathcal{B}_\mathcal{K} \cup \mathcal{B}_\mathcal{N}} \mathcal{L}_{obj}(o_i, o_i^*). \tag{3}$$

Compared with (2), the anchors that overlap with the pseudo boxes of the detected novel objects, i.e., $i \in \mathcal{B}_\mathcal{N}$, are also involved in training. The pseudo boxes can be acquired from a single source, i.e., one of the geometric cues, and from both, i.e., pseudo label ensembling. We name our method GOOD-X when using a specific geometric cue X as the pseudo labeling source, whereas GOOD-Both stands for ensembling the pseudo labels from both the depth and normals.

Inspired by previous works in self-training (Xie et al., 2020; Sohn et al., 2020; Xu et al., 2021), we use strong data augmentation during Phase-II to counteract the noise in pseudo boxes and further boost the performance of GOOD. Specifically, for Phase-II training, we use AutoAugment (Cubuk et al., 2019) which includes random resizing, flip, and cropping.

## 4 Experiments

**Benchmarks.** We target two major challenges of open-world class-agnostic object detection: *cross-category and cross-dataset generalization*. For the cross-category evaluation, we follow the common practice in the literature (Kim et al., 2021; Wang et al., 2022) to split the class category list into two

parts. One is used as the base class for training, whereas the other is reserved only for testing cross-category generalization. Specifically, we adopt two splits of the 80 classes in the COCO dataset (Lin et al., 2014). The first benchmark splits the COCO classes into a single "person" class and 79 non-person classes. This is to stress-test the generalization ability of the methods. We follow Wang et al. (2022) to choose the "person" category as the training class because it contains diverse viewpoints and shapes. The second benchmark splits the COCO classes into 20 PASCAL-VOC (Everingham et al., 2010) classes for training and the other 60 for testing. For the cross-dataset evaluation, we use the ADE20K dataset (Zhou et al., 2019) for testing. We compare models trained using only 20 PASCAL-VOC classes or all 80 COCO classes on detecting objects in ADE20K. This is to evaluate open-world class-agnostic object detectors when used in the wild.

**Implementation.** We use the same architecture as OLN in (Kim et al., 2021) for both Phase I and Phase II training. OLN is built on top of a standard Faster RCNN (Ren et al., 2015) architecture with a ResNet-50 backbone pretrained on ImageNet (Deng et al., 2009). We implement our method using MMDetection framework (Chen et al., 2019) and use the SGD optimizer with an initial learning rate of $0.01$ and batch size of $16$. The models with data augmentation are all trained for 16 epochs. Other models are trained for 8 epochs. We did not find training for longer epochs beneficial for models without data augmentation. The optimal number of pseudo boxes, i.e., $k \in \{1, 2, 3\}$, varies across input types and is determined on a small holdout validation set. See Appendix A for further details.

**Evaluation metrics.** Following (Kim et al., 2021; Wang et al., 2022), we use Average Recall (AR@k) over multiple IoU thresholds (0.5:0.95), and set the detection budget k as $100$ by default. All ARs are shown in percentage. $AR_A$ and $AR_N$ respectively denote the AR score on detecting all classes (including the base and novel ones) and on detecting the novel classes. To evaluate $AR_N$, we do not count the boxes associated to the base classes into the budget k. The same protocol is applied when evaluating per-class ARs and ARs for small, medium and large size of objects, i.e., $AR^{s/m/l}$.

## 4.1 Detecting unknown objects in an open world

Table 1a and 1b compare open-world class-agnostic object detectors on two cross-category benchmarks and two cross-dataset benchmarks, respectively. Our method GOOD, which incorporates geometric cues, considerably outperforms state-of-the-art RGB-based open-world class-agnostic object detection methods, i.e., OLN (Kim et al., 2021) and GGN (Wang et al., 2022). OLN did not involve any novel object into training. Although GGN proposed to use an intermediate pairwise affinity (PA) representation for pseudo labeling, the PA predictor is still trained on the RGB images, therefore it still has the bias towards the known classes as other RGB-based methods.

On the cross-category benchmarks shown in Table 1a, with a single training class "person" on the COCO dataset, GOOD can surpass SOTA methods by 5.0% $AR_N$@100 on detecting objects from non-person classes, a relative improvement of 24%. With 20 PASCAL-VOC classes for training, GOOD surpasses SOTA methods by 6.1% $AR_N$@100 in detecting non-VOC classes, a relative improvement over 18%. On the cross-dataset benchmarks, Table 1b shows that GOOD achieves 2.4% to 4.7% gain on $AR_A$@100 in different setups. We observe that GOOD is particularly strong when there are fewer training classes, i.e., person → non-person and VOC → ADE20k. For RGB-based methods, the overfitting problems become more severe as the object diversity from the base classes reduces. In contrast, the geometric cues can still detect novel-looking objects, which are particularly helpful to training with only a limited number of base class annotations. Finally, ensembling both geometric cues offers additional performance gains.

## 4.2 Advantages of geometric cues

**Geometric cues are less sensitive to appearance shifts across classes.** We first compare per-novel-class AR@5 of the object proposal network trained on geometric cues with that trained on the RGB image. Here, AR@5 is of interest as geometric cues are used to discover novel-looking objects during Phase-I and no more than five pseudo boxes per image will be used in Phase-II, see Figure 3.

Figure 4 shows that geometric cues can achieve much higher per-novel-class AR@5 than RGB in many categories. An example is the novel supercategory "food", including classes such as "hot dog" and "sandwich". The base classes, belonging to the supercategory "person", "animal", "vehicle", and "indoor", have very different appearances to the "food" supercategory. The RGB-based model

| | Person→Non-Person | | | | | VOC→Non-VOC | | | | |
|---|---|---|---|---|---|---|---|---|---|---|
| Method | $AR_A$ | $AR_N$ | $AR_N^s$ | $AR_N^m$ | $AR_N^l$ | $AR_A$ | $AR_N$ | $AR_N^s$ | $AR_N^m$ | $AR_N^l$ |
| FRCNN (oracle) | 55.9 | 53.4 | 37.9 | 59.5 | 73.0 | 55.9 | 52.6 | 37.1 | 60.0 | 73.1 |
| FRCNN (cls-agn) | 25.7 | 12.2 | 8.7 | 12.4 | 18.2 | 38.5 | 27.3 | 10.8 | 30.2 | 55.8 |
| OLN (Kim et al., 2021) | 30.9 | 16.5 | 8.7 | 14.7 | 33.4 | 47.5 | 33.2 | 18.7 | 39.3 | 58.6 |
| GGN (Wang et al., 2022) | 30.3 | 20.7 | 12.0 | 25.6 | 29.6 | 39.8 | 31.5 | 11.8 | 37.4 | **63.8** |
| SelfTrain-RGB | 32.5 | 18.7 | 11.3 | 18.6 | 32.6 | 48.1 | 37.4 | **22.8** | 43.9 | 57.7 |
| GOOD-Depth | 37.0 | 25.6 | 12.8 | 30.4 | **42.4** | 49.6 | 39.0 | 21.1 | 47.5 | 63.2 |
| GOOD-Normal | 35.6 | 23.7 | 13.9 | 27.6 | 36.0 | 49.6 | 38.9 | 21.2 | 47.9 | 62.0 |
| GOOD-Both | **37.3** | **25.9** | **14.2** | **32.6** | 38.9 | 49.5 | **39.3** | 21.6 | **48.2** | 62.4 |

(a) **Cross-category benchmarks**

| | VOC→ADE20K | | | | COCO→ADE20K | | | |
|---|---|---|---|---|---|---|---|---|
| Method | $AR_A$ | $AR_A^s$ | $AR_A^m$ | $AR_A^l$ | $AR_A$ | $AR_A^s$ | $AR_A^m$ | $AR_A^l$ |
| FRCNN (cls-agn) | 22.6 | 15.5 | 23.7 | 26.5 | 25.9 | 20.5 | 28.5 | 27.4 |
| OLN (Kim et al., 2021) | 29.2 | 19.7 | 30.7 | 34.4 | 32.9 | 25.1 | 35.9 | 35.6 |
| GGN (Wang et al., 2022) | 27.0 | 16.9 | 27.5 | 33.6 | 29.8 | 18.9 | 29.1 | 38.2 |
| SelfTrain-RGB | 27.7 | 18.4 | 29.7 | 32.4 | 33.8 | **26.5** | 37.6 | 35.4 |
| GOOD-Depth | 33.9 | 21.1 | 35.9 | 41.2 | **35.3** | 25.7 | 38.0 | 39.6 |
| GOOD-Normal | 33.4 | 22.0 | 36.2 | 38.8 | 33.5 | 25.7 | 37.1 | 35.5 |
| GOOD-Both | **34.0** | 21.9 | **37.0** | **39.9** | **35.3** | 25.1 | 38.2 | **39.9** |

(b) **Cross-dataset benchmarks**

Table 1: **Detecting unknown objects in an open world.** FRCNN (oracle) is a standard Faster R-CNN detector trained on all COCO classes and serves as a performance upper bound on the cross-category benchmarks. FRCNN (cls-agn) is a Faster R-CNN trained in a class-agnostic manner, serving as a baseline for comparison. Our geometry-guided methods GOOD-Xs are compared with SOTA open-world class-agnostic object detectors, i.e., OLN and GGN. SelfTrain-RGB is RGB-based self-training, i.e., using the RGB image instead of geometric cues for pseudo labeling. We only report $AR_A$ in b) as the classes in ADE20K do not exactly match those in COCO.

has difficulty detecting food using appearance cues. In contrast, the geometric cues can generalize across supercategories. For categories that geometric cues are worse than RGB, we find that they typically are of small sizes, such as knives, forks, and clocks. This shows that while abstracting away appearance details, geometric cues may also lose some information about small objects. This again shows complementariness of RGB and geometric cues.

**Geometric cues are better than edges and pairwise affinities.** We further compare the geometric cues with two other mid-level representations: 2D edge and PA. The 2D edge map is extracted using the Holistically-nested Edge Detection (HED) model (Xie & Tu, 2015), which shows more robust performance across the datasets than more recent methods. The HED model is trained on the Berkeley Segmentation Dataset and Benchmark (BSDS500) dataset (Arbelaez et al., 2011) which contains 500 images with segmentation annotations. PA can be thought of as a learned object boundary predictor. Wang et al. (2022) trained it directly on RGB images and then grouped the predictions into object masks using a combination of traditional grouping methods (Shi & Malik, 2000; Arbeláez, 2006; Arbeláez et al., 2014). We compare these four data modalities as the source of pseudo labeling. From Table 2, we can see that depth and normals outperform 2D edge and PA on detecting novel objects. We speculate that this is because 2D edge and PA mainly capture object boundaries, whereas depth and normals have an extra spatial understanding of the scene and can thus better detect objects in complex scenes. Owing to their better pseudo labels, GOOD-Depth/Normal outperform GOOD-Edge/PA on the final $AR_N$@100, i.e., 39.0%/38.9% vs. 38.1%/37.1%, see Appendix D.

**Geometric cues are better than adding shape bias and multi-scale augmentation.** In the previous part, we showed that geometric cues can detect novel objects from very different supercategories, owing to their less sensitivity to detailed appearance changes in objects such as textures. It is thus natural to compare with RGB-based model with inductive shape bias to counteract the texture bias. We adopt the stylized ImageNet pretrained ResNet-50 from (Geirhos et al., 2019) as the backbone for SelfTrain-RGB. This backbone is trained using heavy style-based augmentation to mitigate the

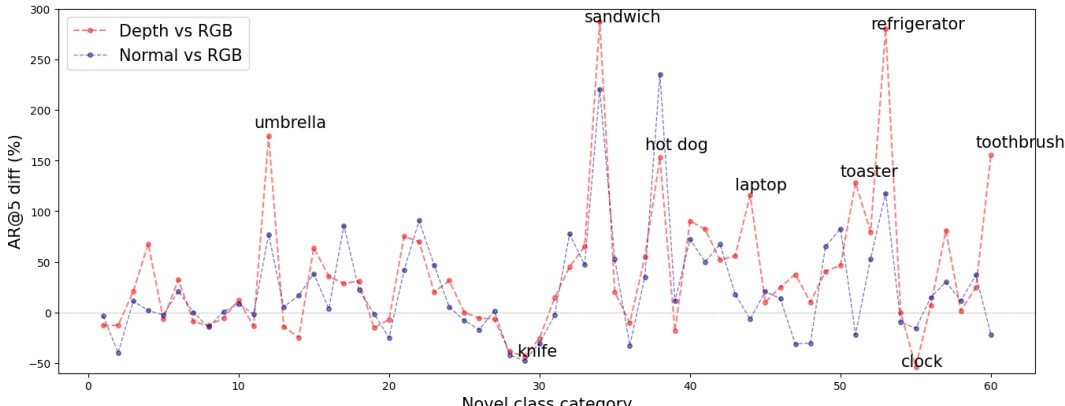

Figure 4: **Per-novel-class AR@5 difference comparison of pseudo boxes on COCO VOC →**
**Non-VOC**. We train the object proposal network on the geometric cues (Phase-I in Figure 3) and
also directly on the RGB image. We show their per-novel-class AR@5 differences, which is defined
as $(AR_X - AR_{RGB})/AR_{RGB}$ with $X \in \{Depth, Normal\}$. The geometric cues outperform the RGB
image on those classes above the zero difference line. We also highlight some classes where RGB
and geometric cues have big differences.

| Modality | $AR_N$ @ 1 | $AR_N$ @ 5 | $AR_N^s$ @ 5 | $AR_N^m$ @ 5 | $AR_N^l$ @ 5 |
|---|---|---|---|---|---|
| RGB | 4.0 | 12.6 | **5.4** | 16.4 | 21.8 |
| Depth | **4.7** | **13.2** | 2.4 | 16.0 | **31.6** |
| Normal | 4.3 | 12.8 | 3.0 | **16.6** | 27.7 |
| Edge | 3.2 | 10.4 | 3.5 | 14.5 | 18.5 |
| Pairwise affinity (PA) | 3.2 | 8.1 | 0.5 | 9.1 | 30.6 |

Table 2: **Comparison on different data modalities for pseudo labeling**. We report $AR_N$@k
achieved by the object proposal network trained on different modalities in Phase-I, where the benchmark is VOC → Non-VOC. Geometric cues (depth and normal) are stronger in discovering novel
objects than the edge and pairwise affinity (Wang et al., 2022).

texture bias, where texture is one of the most discussed appearance features prone to overfit by
RGB-based models. It showed performance improvements on the COCO object detection benchmark under the closed-world assumption (Geirhos et al., 2019). Moreover, we observe from Figure 4
that RGB is relatively stronger in detecting smaller objects. This leads to the question of whether
augmenting SelfTrain-RGB with pseudo boxes extracted at different input scales can already help
this RGB-based method achieve comparable performance to those incorporating geometric cues.

From Table 3, we can see that while the shape bias backbone can improve the $AR_A$ of SelfTrain-RGB, it degrades the performance on novel classes, i.e., $AR_N$, indicating that shape bias obtained
from training on ImageNet may not be very helpful in generalizing to novel objects. As for the
multi-scale augmentation, although it can improve the detection recall of medium and large objects,
the overall performance is still worse than our models that incorporate geometry cues. Overall, these
comparisons show that geometry provides strong cues for object localization.

| Method | $AR_A$ | $AR_N$ | $AR_N^s$ | $AR_N^m$ | $AR_N^l$ |
|---|---|---|---|---|---|
| SelfTrain-RGB | 48.1 | 37.4 | **22.8** | 43.9 | 57.7 |
| ShapeBias | 48.8 | 36.5 | 21.4 | 42.5 | 58.6 |
| ScaleAug | 48.5 | 37.9 | 21.2 | 44.7 | 62.2 |
| GOOD-Both | **49.5** | **39.3** | 21.6 | **48.2** | **62.4** |

Table 3: **Comparison of GOOD and two other strategies on VOC to non-VOC**. ShapeBias replaced the backbone of SelfTrain-RGB with a stylized ImageNet pre-trained backbone. ScaleAug
applied multi-scale augmentation to the RGB input for collecting pseudo boxes at different scales.

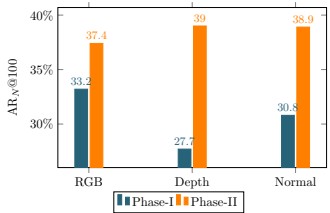 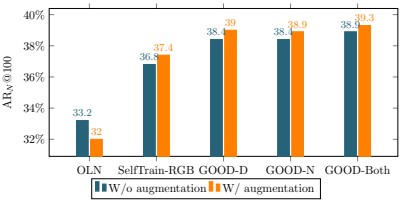 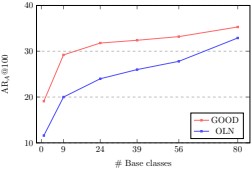

(a) Effectiveness of pseudo labeling.  (b) Effectiveness of data augmentation.  (c) Varying the number of base classes.

Figure 5: **Ablation studies.** (a) and (b) are conducted on COCO VOC to non-VOC benchmark. The study on the number of base classes (c) uses ADE20K as the evaluation benchmark.

| Supercategories | Person | +Vehicle | +Outdoor, Animal | +Accessory, Sports | +Kitchen, Food | +Furniture, Electronic, Appliance, Indoor |
|---|---|---|---|---|---|---|
| # Training classes | 1 | 9 | 24 | 39 | 56 | 80 |
| # Training images | 64115 | 74152 | 92169 | 93939 | 107036 | 117266 |
| # Training instances | 609666 | 654460 | 715582 | 727207 | 824535 | 860001 |

Table 4: **Base class choices for studying the influence of the number of base classes.**

## 4.3 ABLATION STUDIES

In this section, we conduct ablation studies to understand the effectiveness of pseudo-labeling, data augmentation, and the influence of the number of base classes. Figure 5a) shows the $AR_N@100$ achieved by the object proposal network (Phase-I) and the object detector (Phase-II). The latter uses the pseudo boxes generated by the former, where the former can be trained with different data modalities. The RGB-based object proposal network outperforms the depth- and normal-based one on $AR_N@100$. This indicates that depth and normal maps cannot simply replace RGB image for object detection. As evidenced in Table 2 and Figure 4, they are competitive on $AR_N@k$ with $k \leq 5$, meaning their top predictions are of high quality. Pseudo labeling is thus an effective way for geometry-guided open-world object detector training.

We further study the effect of AutoAugment (Cubuk et al., 2019). Using it during Phase-II training, we achieve higher $AR_N@100$ as shown in Figure 5b). Data augmentation is helpful to counteract the noise in pseudo labels. However, using AutoAugment to train OLN on the ground truth base class annotations, we observe only improvement in recalling the base class objects (from 58.4% to 61.7% $AR_{Base}@100$), but degradation in novel object detection. This shows that data augmentation via random resizing, cropping, and flipping cannot improve generalization across categories.

Finally, we study the influence of the number of base classes. As shown in Table 4, we split the COCO dataset based on the supercategories to create six different splits. We then train GOOD and OLN on these splits and evaluate them on ADE20K. More base classes in training allow object detectors to learn a more generic sense of objectness so that they can better detect novel objects. As shown in Figure 5c, both GOOD and OLN achieve better $AR_N@100$ along with the number of base classes, and their performance gap reduces. However, the annotation cost also grows along with the number of classes. GOOD can achieve a similar $AR_N@100$ as OLN with only half the number of base classes, e.g., 39 vs. 80. This shows that GOOD is more effective at learning a generic sense of objectness and less prone to overfitting to the base classes.

## 5 CONCLUSION

In this paper, we proposed a method GOOD for tackling the challenging problem of open-world class-agnostic object detection. It exploits the easy-to-obtain geometric cues such as depth and normals for detecting unannotated novel objects in the training set. As the geometric cues focus on object shapes and relative spatial locations, they can detect novel objects that RGB-based methods cannot detect. By further involving these novel objects into RGB-based object detector training, GOOD demonstrates strong generalization performance across categories as well as datasets.

ACKNOWLEDGMENT

Haiwen Huang and Dan Zhang were supported by Bosch Industry on Campus Lab at University of Tübingen. Andreas Geiger was supported by the ERC Starting Grant LEGO-3D (850533) and the DFG EXC number 2064/1 - project number 390727645. Haiwen Huang would like to thank Tianlin Ye for her emotional support throughout the project and Jojumon Kavalan for generously providing access to his Internet during the rebuttal period.

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

| Method | LVIS COCO→Non-COCO | | | | | COCO→UVO | | | |
|---|---|---|---|---|---|---|---|---|---|
| | $AR_A$ | $AR_N$ | $AR_N^s$ | $AR_N^m$ | $AR_N^l$ | $AR_A$ | $AR_A^s$ | $AR_A^m$ | $AR_A^l$ |
| FRCNN (cls-agn) | 26.6 | 21.0 | 14.9 | 32.7 | 36.2 | 42.3 | 22.2 | 38.3 | 52.0 |
| OLN (Kim et al., 2021) | 32.2 | 27.4 | 17.9 | 44.7 | 53.1 | 49.2 | 35.0 | 48.7 | 55.1 |
| GGN (Wang et al., 2022) | 27.1 | 22.5 | 15.7 | 35.5 | 38.4 | 45.6 | 25.6 | 43.2 | 54.9 |
| SelfTrain-RGB | 32.6 | 28.3 | 19.0 | 45.7 | 52.2 | 48.7 | 35.8 | 48.8 | 53.5 |
| GOOD-Depth | 32.8 | 28.3 | 18.3 | 46.8 | **54.1** | **50.3** | 35.6 | 49.9 | **56.4** |
| GOOD-Normal | **33.4** | **29.2** | **19.3** | **49.8** | 53.3 | 49.8 | 35.8 | 50.0 | 55.0 |
| GOOD-Both | 33.2 | 29.0 | 19.0 | 47.7 | 53.0 | **50.3** | **36.1** | **50.2** | 55.4 |

Table 5: **More benchmarks.** The same methods in Table 1 are compared.

# A   IMPLEMENTATION DETAILS

GOOD uses OLN (Kim et al., 2021) as the architecture for both Phase-I and Phase-II training. OLN is built on top of Faster RCNN (Ren et al., 2015). For open-world object detection, the classification heads are replaced with the objectness score prediction heads, i.e., predicting the centerness and IoU of each bounding box proposal at the two stages, respectively. We use the objectness score $\sqrt{\text{centerness} \times \text{IoU}}$ for ranking the pseudo boxes and selecting the top $k$ pseudo boxes per image. The optimal $k$ choice for GOOD-Depth and GOOD-Normal is 1 and for SelfTrain-RGB is 3.

For Phase-I training, we trained the proposal network with loss as given in Eq. 2 and used the SGD optimizer with an initial learning rate of 0.01 and batch size of 16 for 8 epochs. For Phase-II training, unless otherwise stated, we trained the object detector with loss as given in Eq. 3 and used SGD optimizer with an initial learning rate of 0.01 and batch size of 16 for 16 epochs with AutoAugment. For GOOD-Both, we merge the pseudo boxes generated by object proposal networks separately trained on depth and normal maps by filtering out the overlapping boxes. Specifically, if the IoU of two pseudo boxes is larger than 0.5, they are seen as overlapping with each other, and the one with lower objectness score will be filtered out. For other ensembling experiments, if not specified, pseudo boxes are also merged as described above.

We use the DPT-Hybrid models from Omnidata repository (Eftekhar et al., 2021) for off-the-shelf inference of geometric cues. The DPT-Hybrid models (Ranftl et al., 2021a) have a hybrid architecture of attention layers and convolutional layers. They are trained on the Omnidata Starter Dataset (Eftekhar et al., 2021) for one week with 2D and 3D data augmentations (Kar et al., 2022), and one week with cross-task consistency (Zamir et al., 2020) on 4 V100 GPUs. To infer on RGB images, we first pad images to sizes divisible by 32 without resizing, then feed them to the DPT-Hybrid model. Note although the original models are trained on 384×384 image patches, we find that inferring on the original resolution of COCO produces better visual results than on the resolution of 384×384.

# B   MORE BENCHMARKS

We further evaluate our approach on more benchmarks. Specifically, we evaluated the baselines and GOOD on the cross-category benchmark LVIS COCO to non-COCO and cross-dataset benchmark COCO to UVO. The results are shown in Table 5. As expected, the performance gains are smaller than those observsed on benchmarks with smaller number of base classes such as COCO VOC to non-VOC. But still, GOOD surpasses all the state-of-the-art RGB-based methods. This shows that even in extreme conditions when the number of base classes is large, GOOD can still be helpful in improving the open-world performance.

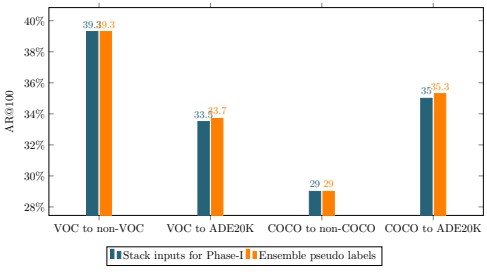

(a) Choice of ensembling geometric cues.

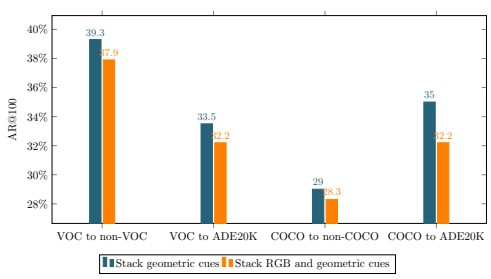

(b) Fusing inputs: whether to use RGB.

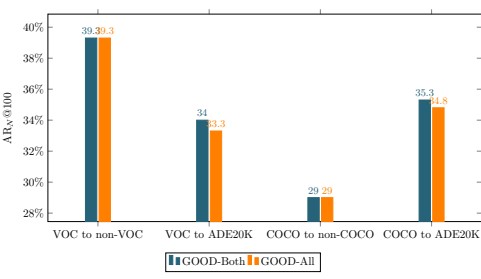

(c) Ensemble pseudo boxes: whether to use RGB.

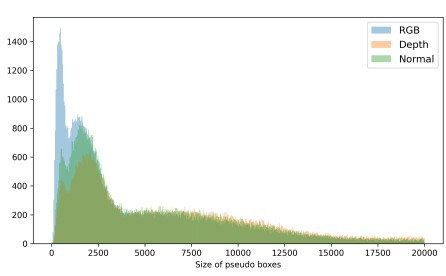

(d) Histograms of top1 pseudo box sizes from models trained on COCO VOC with different modalities.

Figure 6: **More ablation studies.**

## C  MORE ABLATION STUDIES

### C.1  ENSEMBLING WAYS OF GEOMETRIC CUES

There are two possible ways to ensemble geometric cues: (1) Stack the two geometric cues together and train a single object proposal network on these stacked inputs in Phase-I; (2) Train two object proposal networks and extract pseudo boxes separately, then merge them into a single pseudo box pool for Phase-II training. The details of the merging process is described in Appendix A. We conduct ablation studies on these two methods. From Figure 6a, we demonstrate that empirically, ensembling pseudo labels is slightly better than using stacked inputs for Phase-I training. Throughout the paper, we use the pseudo label ensembling for GOOD-Both.

### C.2  ON INCORPORATING RGB IN PHASE-I

It is natural to think of incorporating RGB in Phase-I. To do so, we can also consider the two ways for ensembling geometric cues. If we stack RGB with geometric cues to train the proposal network, the model will tend to make more use of RGB to optimize the target localization loss. This is because RGB is a much stronger input signal than geometric cues in the closed-world setup — $AR@100_{base}$ is 58.3 for RGB inputs alone and 44.9 when stacking depth and normals on COCO VOC classes. In the extreme case, the model can even completely ignore geometric cues. This reliance on RGB inputs prevents the model from making the best use of geometric cues to discover novel objects in Phase-I, which is crucial for open-world object detection. As shown in Figure 6b, stacking RGB with geometric cues leads to inferior performance across many benchmarks.

Alternatively, we can train a separate object proposal on RGB inputs, extract pseudo boxes, and merge them with those extracted from models trained on geometric cues. We can name this method "GOOD-All". In Figure 6c, we compare GOOD-All with GOOD-Both and find that adding pseudo boxes from RGB (i.e., GOOD-All) either leads to no performance gains or even worsens the performance on benchmarks like VOC to ADE20K. To understand this, we note that object proposal networks trained on RGB favor smaller detection boxes, as evidenced in our visualizations (Figure 10, 11) and more quantitatively in histograms (Figure 6d). These smaller detection boxes can either be small objects or just textures and parts of larger objects, which can potentially hurt the

| Method | VOC→Non-VOC | | | | | VOC→ADE20K | | | |
|---|---|---|---|---|---|---|---|---|---|
| | $AR_A$ | $AR_N$ | $AR_N^s$ | $AR_N^m$ | $AR_N^l$ | $AR_A$ | $AR_A^s$ | $AR_A^m$ | $AR_A^l$ |
| GOOD-Both | **49.5** | **39.3** | 21.6 | **48.2** | **62.4** | **34.0** | 21.9 | **37.0** | **39.9** |
| GOOD-All | 48.5 | **39.3** | **22.7** | 47.6 | 60.8 | 33.3 | **22.9** | 36.2 | 38.0 |

Table 6: **More comparison of GOOD-Both and GOOD-All.** For GOOD-All, the performance gains in detecting small objects ($AR^s$) are too small to compensate for the losses in detecting larger objects ($AR^m$ and ($AR^l$)), leading to overall inferior performance.

| Method | VOC→Non-VOC | | | | | VOC→ADE20K | | | |
|---|---|---|---|---|---|---|---|---|---|
| | $AR_A$ | $AR_N$ | $AR_N^s$ | $AR_N^m$ | $AR_N^l$ | $AR_A$ | $AR_A^s$ | $AR_A^m$ | $AR_A^l$ |
| FCOS | 43.6 | 29.3 | 14.6 | 35.7 | 50.2 | 25.6 | 17.2 | 26.5 | 30.9 |
| OLN | 47.5 | 33.2 | 18.7 | 39.3 | 58.6 | 29.2 | 19.7 | 30.7 | 34.4 |
| GOOD-Both (FCOS) | 48.4 | 36.3 | 18.4 | 46.5 | 58.0 | 32.5 | 20.1 | 34.5 | 39.5 |
| GOOD-Both (OLN) | 49.5 | 39.3 | 21.6 | 48.2 | 62.4 | 34.0 | 21.9 | 37.0 | 39.9 |

Table 7: **Architecture choice.** FCOS is a single-stage proposal-free object detector, and OLN is a two-stage proposal-based object detector (modified from Faster R-CNN). GOOD can significantly improve the open-world performance of both architectures.

performance of the final detector to detect large objects. This is consistent with our observations in Table 6. Compared to GOOD-Both, the gains in $AR_{N(A)}^s$ are usually too small to compensate for the losses in $AR_{N(A)}^l$, leading to the inferior overall performance of GOOD-All.

## C.3 ARCHITECTURE CHOICE

In principle, our approach is model agnostic and is therefore compatible with both proposal-free and proposal-based object detectors. To demonstrate this, besides the two-stage proposal-based detectors in the main paper, we also experiment with a more recent single-stage proposal-free object detector FCOS (Tian et al., 2019). Specifically, we kept Phase I unchanged, i.e., generating the geometric cue-based pseudo boxes using OLN as the architecture. In Phase II, we train a class-agnostic FCOS using the extracted pseudo boxes together with the groundtruth annotations of the base classes. The experiment results are shown in Table 7. We can see that GOOD can significantly improve FCOS in terms of detecting novel objects and OLN is a stronger architecture than FCOS to be used in GOOD.

## D MORE DISCUSSION ON DIFFERENT MODALITIES FOR GOOD

In this section, we further discuss how different modalities behave and can be further ensembled to boost the performance. We first compare different modalities used for Phase-I training and pseudo labeling in GOOD on the COCO VOC to non-VOC benchmark. In Table 8, we show that pseudo-labeling using the geometric cues leads to stronger performances. This agrees with our study in Table 2 where we found that pseudo boxes from proposal networks trained on geometric cues have higher $AR_N@5$, indicating that they are of higher quality.

## D.1 COMPLEMENTARINESS OF DIFFERENT MODALITIES

In the main paper, we have combined the pseudo boxes only from the geometric cues, i.e., depth and normals. GOOD-Both provides additional performance gains over GOOD-Depth and GOOD-Normal. As we have more than two sources of pseudo labeling, it is natural to examine further if they are complementary and thus can be jointly exploited. We first evaluate the overlap of their top-1 ranked pseudo boxes. Their most confident novel object detections best convey their bias in generalization. We can observe from Figure 7a that the overlap is low across all the input types. Table 2 and Table 8 further reveal their complementariness in detecting different sizes of objects. Both

| Modality | $AR_N$ | $AR_N^s$ | $AR_N^m$ | $AR_N^l$ |
|---|---|---|---|---|
| SelfTrain-RGB | 37.4 | **22.8** | 43.9 | 57.7 |
| GOOD-Edge | 38.1 | 21.8 | 45.7 | 60.2 |
| GOOD-PA | 37.1 | 18.6 | 43.9 | **65.3** |
| GOOD-Depth | **39.0** | 21.1 | 47.5 | 63.2 |
| GOOD-Normal | 38.9 | 21.4 | **47.9** | 62.4 |

Table 8: **Comparison of GOOD using different modalities on COCO VOC to non-VOC benchmark.**

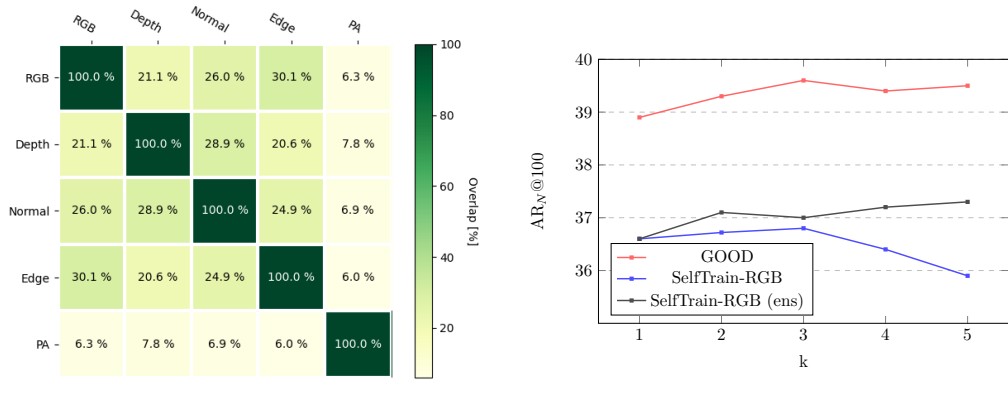

(a) Overlap of top1 pseudo boxes.

(b) Influence of top-k when ensembling.

Figure 7: **Complementariness of different representations.** In (b), **SelfTrain-RGB** is using the top-k pseudo boxes from a single RGB-based object proposal netowrk for retraining, and **SelfTrain-RGB (ens)** is using pseudo boxes merged from k independently trained RGB-based object proposal netowrks for retraining.

observations motivate us to ensemble different sources of pseudo boxes, exploiting their diversity for training the object detector.

To decide which modality to ensemble first, we designed a simple greedy algorithm based on the overlap of pseudo boxes and the potential performance gain of adding the modality to the ensemble. Specifically, starting with the best-performing modality (depth), we incrementally ensemble more sources of pseudo boxes by selecting the source with the highest $Utility * Uniqueness$ score, where $Utility$ is the performance gain against a vanilla OLN, and $Uniqueness$ is the overlap of the pseudo boxes with the current ensemble pseudo boxes. The performance is evaluated on a holdout validation set. The chosen ensemble order for the COCO VOC to non-VOC benchmark is: depth, normal, PA, edge, RGB.

We show the ensembling results in Figure 7b. Two baselines for using multiple pseudo boxes from RGB are considered: **SelfTrain-RGB** is using the top-k pseudo boxes from a single RGB-based object proposal netowrk for retraining, and **SelfTrain-RGB (ens)** is using pseudo boxes extracted and merged from k independently trained RGB-based object proposal networks for retraining. We can see that ensembling pseudo sources from multiple modalities is better than adding more pseudo boxes from a single source (RGB).

# E    MORE VISUALIZATION

## E.1    VISUALIZATION OF GEOMETRIC CUES

We visualize more examples of geometric cues in Figure 8 and Figure 9. We demonstrate that the inferred geometric cues are of high quality in diverse scenes.

### E.2 VISUALIZATION OF PSEUDO BOXES FROM PHASE-I

We also provide visualization of pseudo boxes in Figure 10 and Figure 11. The pseudo boxes are generated from OLNs trained on RGB, depth, and normals, respectively. We find that pseudo boxes from RGB-based models generally tend to target small objects, textures, and parts of objects. This again shows that RGB-based models over-rely on appearance cues and can overfit to textures and discriminative parts of the training classes.

### E.3 VISUALIZATION OF GOOD DETECTIONS ON NOVEL OBJECTS

We further added more visualization examples of GOOD detection results in Figure 12. The test images contain objects that are seen neither in the GOOD training set (COCO) nor the Omnidata model training set. The presented examples include new technology devices, spaceships, dinosaurs, aliens, and sea scenes. We can see that GOOD can still make reasonable object bounding box predictions even though these objects have never appeared in the training set.

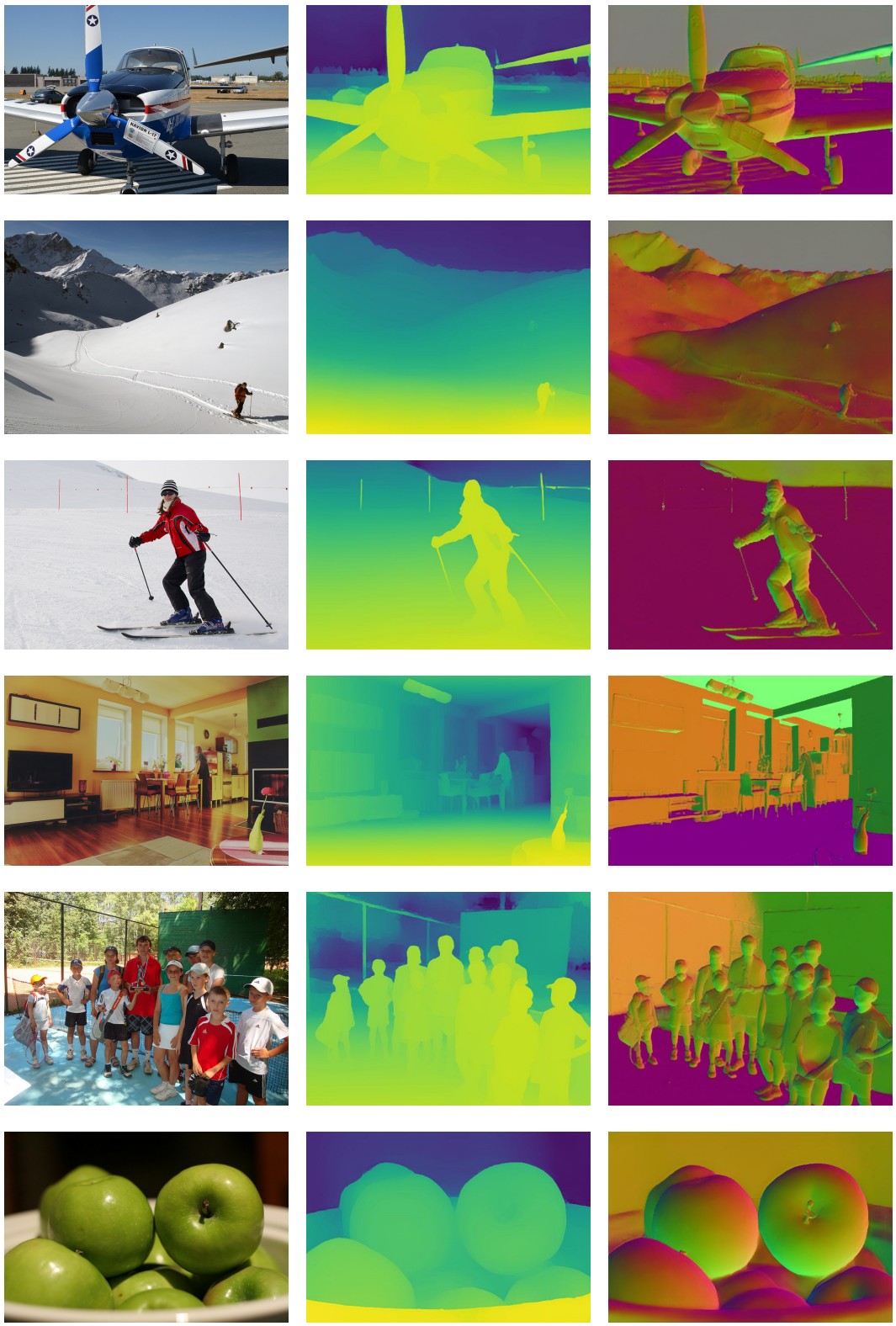

Figure 8: **Visualization of geometric cues.** From left to right: RGB, depth, normals.

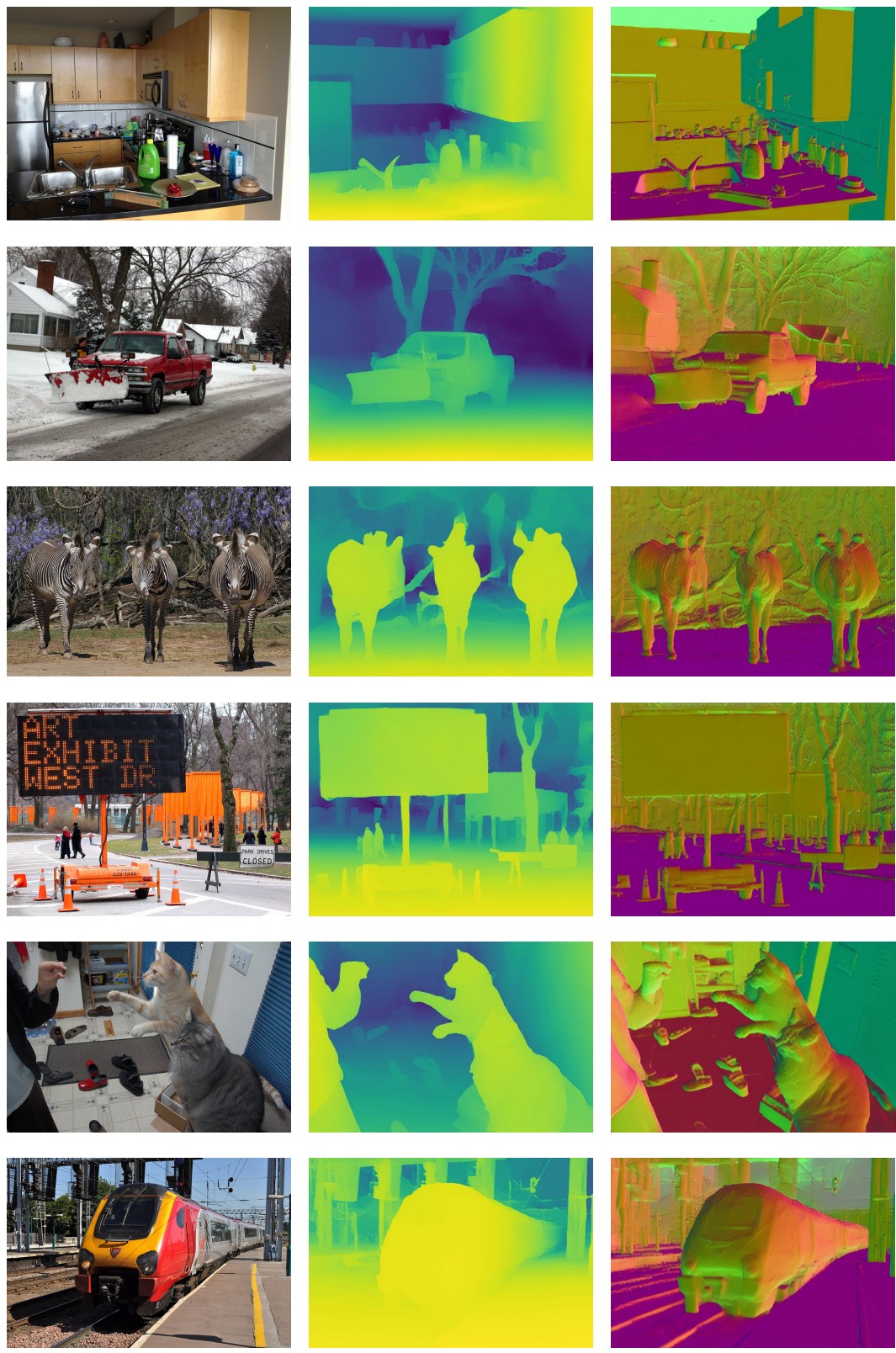

Figure 9: **Visualization of geometric cues.** From left to right: RGB, depth, normals.

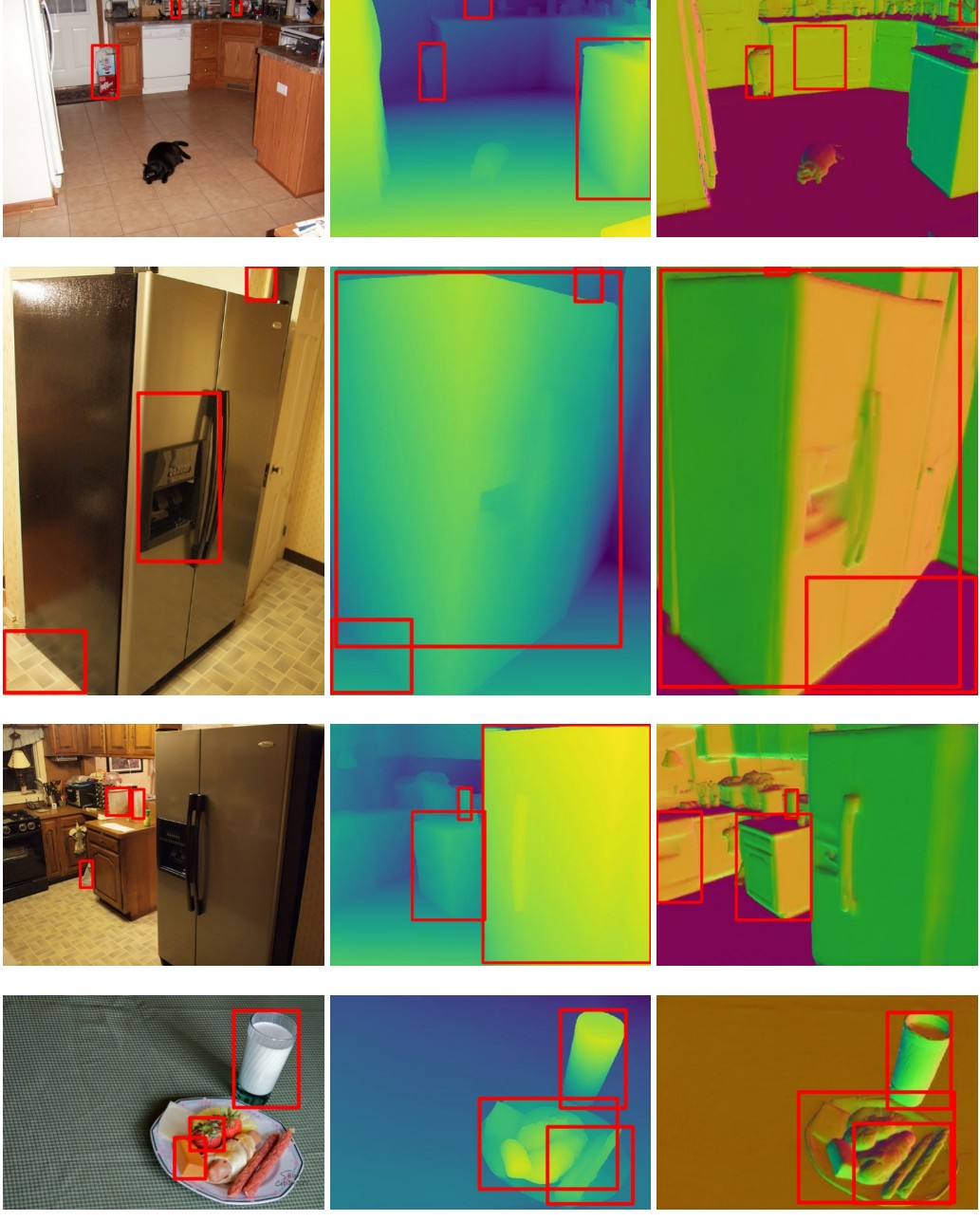

Figure 10: **Top3 pseudo boxes after filtering out those that overlap with known (VOC) class bounding boxes.** The pseudo boxes are generated from OLNs trained on RGB, depth, and normals, respectively. OLN trained on RGB tends to detect small objects and parts of objects.

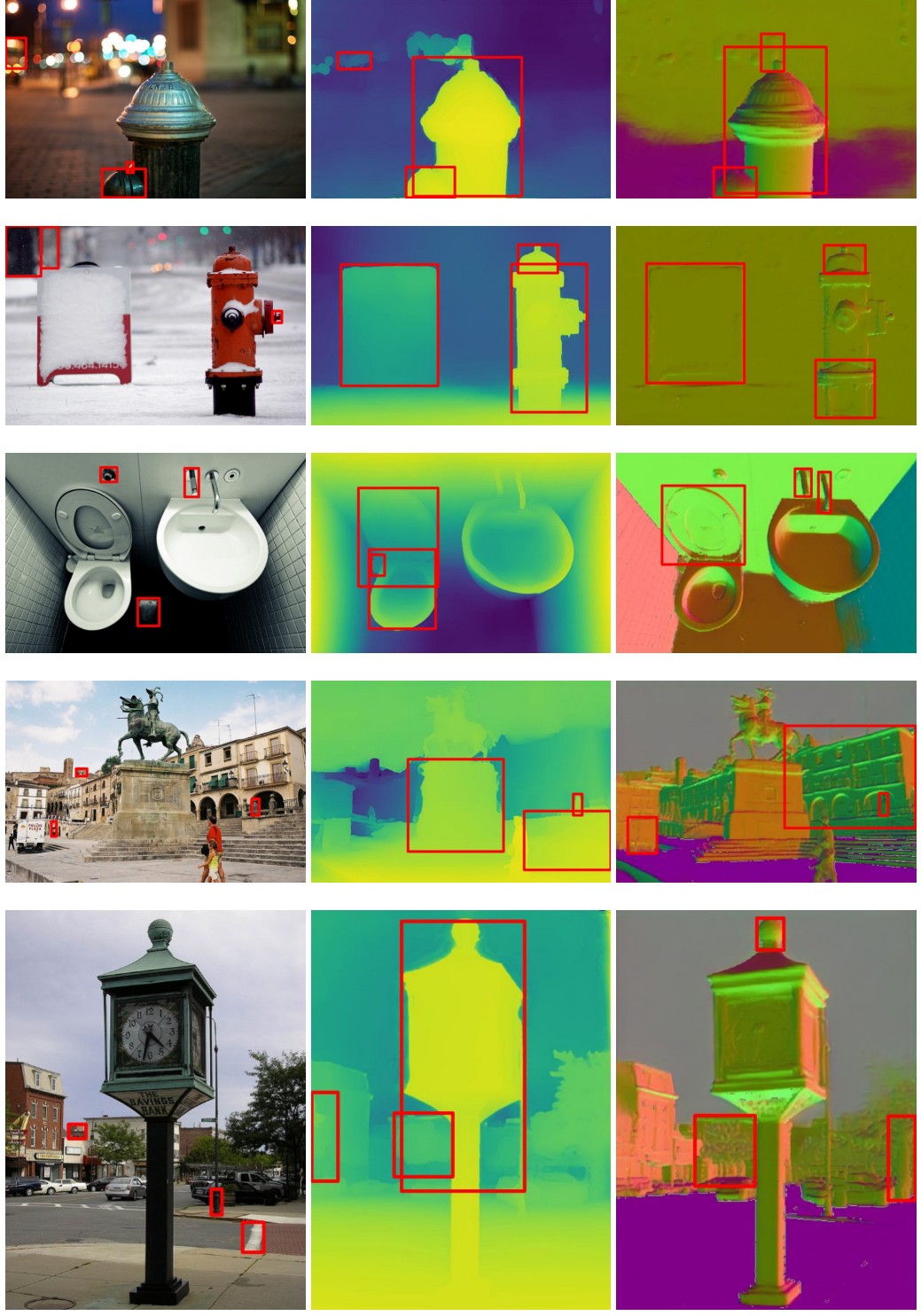

Figure 11: **Top3 pseudo boxes after filtering out those that overlap with known (VOC) class bounding boxes.** The pseudo boxes are generated from OLNs trained on RGB, depth, and normals, respectively. OLN trained on RGB often detect textures or small parts of the objects.

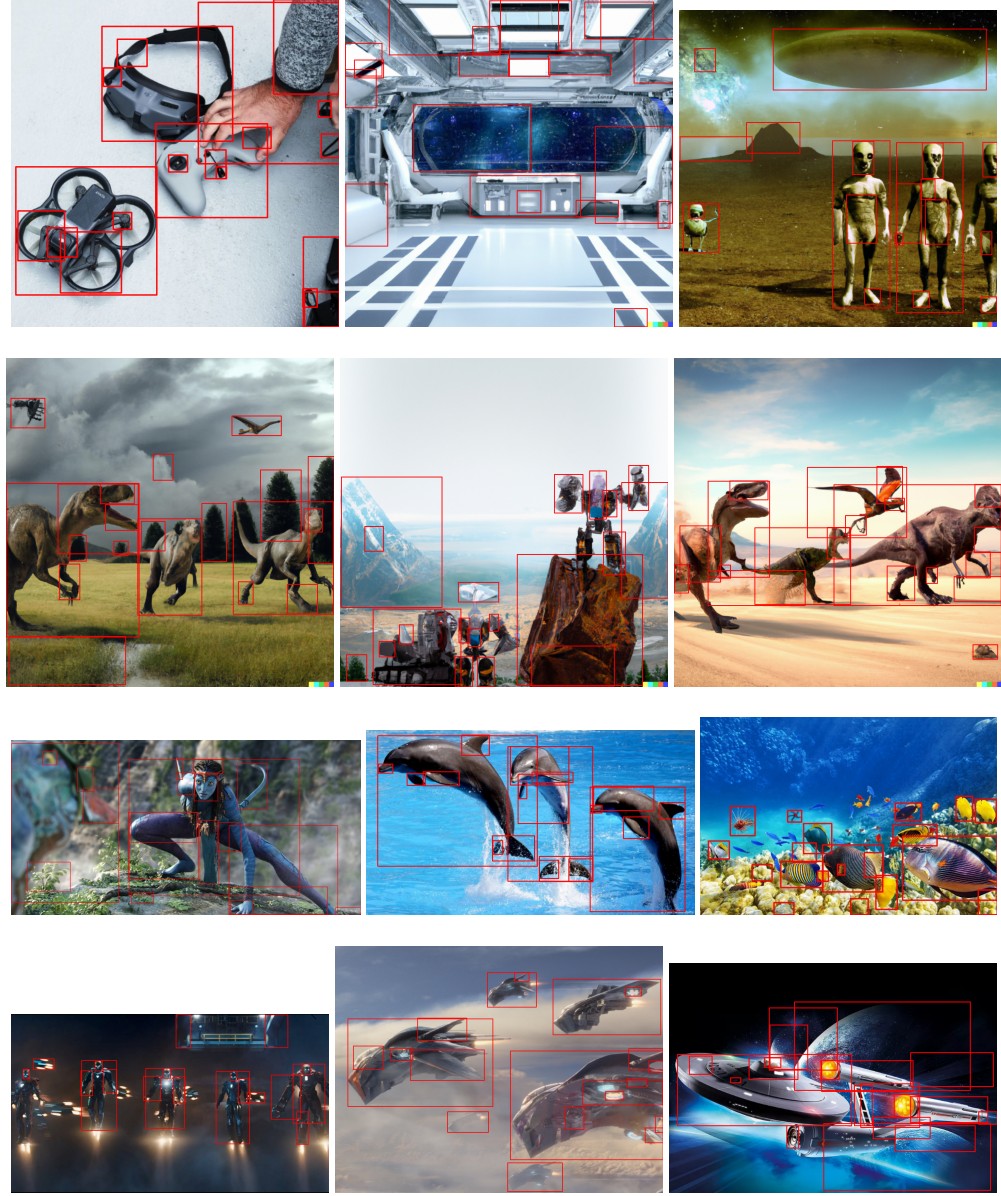

Figure 12: **GOOD detections on novel objects.** Only top 20 detection boxes are shown with the images. The novel objects are seen neither in GOOD training, nor in Omnidata training set.

