# OpenReview forum: "GOOD: Exploring geometric cues for detecting objects in an open world"
_ICLR.cc/2023/Conference — ICLR 2023 poster_

### Official Review · Reviewer_e8x4 · 2022-10-25

**Confidence:** 5
**Correctness:** 3
**Technical Novelty And Significance:** 3
**Empirical Novelty And Significance:** 3
**Recommendation:** 6

**Clarity, Quality, Novelty And Reproducibility:**

This paper clarifies its contribution well and provides details to reproduce the experiments. The novelty is somewhat new.


**Strength And Weaknesses:**

*Strength
1. GOOD reveals that Depth and normal images are superior to edge and PA images for generating pseudo-labels in Table 2. Whereas, it would be better if RGB images are taken as a comparison.

2. The idea to leverage the depth and normal images in the open-world class-agnostic object detection task is simple and innovative. Low-level information deserves more attention in open-world tasks.

3. GOOD achieves state-of-the-art results in several datasets.

*Weaknesses
1. The experiments are insufficient. For example, in cross-dataset benchmarks in Table 1, GOOD only selects ADE2k as the target test set. However, most approaches [1, 2] mainly consider UVO [3] as the target dataset for the cross-dataset benchmark.

2. Heuristically, pseudo-labeling methods work on the partially labeled training set, such as VOC categories in COCO. The effects need to be further validated if the training set is exhaustively (e.g. UVO dataset) or nearly exhaustively (e.g. COCO to Non-COCO in LVIS dataset) labeled. The authors need to add at least one experiment under such scenes.

3. There are several mistakes in the paper, such as the legend in Fig. 7b.

[1] Wang, W., Feiszli, M., Wang, H., Malik, J. & Tran, D. Open-World Instance Segmentation: Exploiting Pseudo Ground Truth From Learned Pairwise Affinity. CVPR (2022).
[2] Saito, K., Hu, P., Darrell, T. & Saenko, K. Learning to Detect Every Thing in an Open World. Arxiv (2021).
[3] Wang, W., Feiszli, M., Wang, H. & Tran, D. Unidentified Video Objects: A Benchmark for Dense, Open-World Segmentation. ICCV (2021).

**Summary Of The Paper:**

The paper focuses on open-world class-agnostic object detection tasks. This paper proposes a pseudo-labeling approach to alleviate the issue that base object categories are limited. Different from conventional methods that acquire pseudo labels from RGB-based images, this paper acquires pseudo labels from depth or normal images. The method, named GOOD, achieves state-of-the-art results on several datasets.

**Summary Of The Review:**

The paper proposes a novel and effective pseudo-labeling method for open-world class-agnostic object detection. The results acquired are superior as well. However, the authors need to conduct the necessary experiments as requested in the weaknesses for a comprehensive comparison. The reviewer will increase the rating if the above concerns are well addressed.

---

> ### Author Response · Authors · 2022-11-17
> **Answer to Reviewer e8x4**
>
> Thanks for your review! We address your concerns as the following:
>
> **1. More benchmarks.**
>
> We have included LVIS as an additional cross-category benchmark and UVO as an additional cross-dataset benchmark. See Table 5 in our revised paper. Our results showed that GOOD has also surpassed previous state-of-the-art methods on these two benchmarks, e.g., GOOD improves AR_N from 27.4 (OLN) to 29.2 on the LVIS COCO to non-COCO benchmark, and improves AR_A from 49.2 (OLN) to 50.3 on the COCO to UVO benchmark.
>
> **2. On the scenario when the training set is exhaustively or nearly exhaustively annotated.**
>
> If the training set has already been exhaustively or nearly exhaustively annotated, the pseudo-labeling will find it difficult to discover novel objects, making it less effective. We have added experiments on the LVIS benchmark (COCO to non-COCO) in Table 5. As expected, while GOOD still surpassed the SOTA open-world object detectors, the performance gains are smaller on benchmarks like COCO VOC to non-VOC.
>
> There are some further points we want to discuss here:
> - If the training set is exhaustively annotated and also contains a large number of base classes, the chance of meeting objects with novel appearances can be very low during inference, making the task more like “closed-world” setup. In this case, not just our method, but any other open-world object detector, won’t be significantly stronger than standard object detectors. However, to have such a dataset, the data collection and curation effort would also grow tremendously, which is unrealistic for many practical applications. Relatedly, in Figure 5c we show that GOOD can achieve similar performance with much fewer annotated training classes (e.g. 39 vs 80), potentially saving lots of annotation efforts.
> - While pseudo-labeling cannot be used to discover new objects in the training set when the dataset is exhaustively annotated, it can still be used on unannotated datasets. For example, pseudo-labeling is widely used in semi-supervised learning [1,2]. As a potential extension, GOOD can also be used for semi-supervised open-world object detection.
> - Besides pseudo-labeling, geometric cues may still be used in other ways to enhance open-world performance. We hope our work can inspire future research works to explore more diverse ways to incorporate the geometric cues for the open-world and out-of-distribution generalization tasks.
>
>
> **3. Fig. 7b**
>
> Thanks for pointing out the typo! We have corrected it in Fig. 7b. We have also added a baseline, SelfTrain-RGB(ens), which ensembles pseudo boxes extracted from independently trained RGB-based object proposal networks.
>
> **4. RGB baseline Table 2**
>
> We have added RGB as an additional comparison baseline in Table 2.
>
> Thank you again for your comprehensive review (including the appendix) and your valuable suggestions!
>
>
> **References**
>
> [1] Xu, Mengde, et al. "End-to-end semi-supervised object detection with soft teacher." Proceedings of the IEEE/CVF International Conference on Computer Vision. 2021.
>
> [2] Li, Gang, et al. "Pseco: Pseudo labeling and consistency training for semi-supervised object detection." European Conference on Computer Vision. Springer, Cham, 2022.

---

> > ### Comment · Reviewer_e8x4 · 2022-12-06
> > **I decide to maintain my rating as weak accept**
> >
> > Most of my concerns have been addressed in the rebuttals.  The GOOD method achieves consistent performance improvements on the COCO->UVO and the COCO->non-COCO settings. The authors have demonstrated a simple yet effective approach for using geometric cues in open-world detection tasks. But the pseudo-labeling baseline is a little weak. Overall, I think the idea of the paper is worth sharing and I decide to maintain my rating as weak accept.

---

> ### Author Response · Authors · 2022-11-30
> **More questions?**
>
> Dear Reviewer e8x4,
>
> Thank you again for your review. We hope our rebuttal can satisfactorily address all your questions and concerns regarding the results on more datasets and the scenarios of exhaustively or near exhaustively annotated training datasets. We wonder if you still have any concerns we could address.
>
> Thank you for your time.

---

### Official Review · Reviewer_3ZGc · 2022-10-25

**Confidence:** 4
**Correctness:** 4
**Technical Novelty And Significance:** 4
**Empirical Novelty And Significance:** 4
**Recommendation:** 8

**Clarity, Quality, Novelty And Reproducibility:**

- Clarity, quality, novelty: no concerns
- Reproducibility: Code release would help significantly. In the absence of this, a table that explicitly states the hyperparams for the teacher (pseudo-labeling) model vs. the student model would be critical. Even if the hyperparams are the same, stating them explicitly in a table would simplify reproduction and address any confusion.


**Strength And Weaknesses:**

Strengths:
- Intuitive, novel method that outperforms prior work on important benchmarks
- Many reasonable ablations that compare against baselines that help show the value of geometric cues vs. self-training in general.
- Good analysis of where the method improves, and where the RGB self-train method falls short or outperforms vs. the proposed method.

Weaknesses:
- Segmentation: One would expect that a key win from dense geometric cues would be improved segmentation of unseen objects. This work (and possibly the results) would be much stronger if evaluated on instance segmentation rather than bounding-box object detection. Admittedly, this is a more complex task, but would increase the impact of this paper.
- As one might expect, the accuracy improvements drop off significantly as the # base classes increases. It might be nice to plot accuracy on novel classes as the # base classes increases (you already have two points for this plot in Table 1 (b) at n=20 and n=80).
- Relatedly, it would have been nice to see experiments on the LVIS dataset, which covers many categories, and has a vocabulary that matches to COCO. This would have allowed authors to report AR_N for LVIS, unlike for ADE20k in Table 1 (b).
- It would have been nice to see how well the model generalizes to objects which are not in the training sets of the depth/normal estimation models. This could even be qualitative, on a few images of objects known to be missing from the original training sets (e.g., a new object from a movie, or a recently introduced tech device).
- Nits:
    - I would highly recommend removing Fig 5, or at least replacing it with 5-10 randomly chosen images, instead of 1 (likely carefully selected) image. It’s difficult to generalize from this one example, and can lead unsuspecting readers to lend excess credence to the conclusion drawn from that example.
    - Fig 7b – (1) do you take more pseudo boxes from a single RGB model? Wouldn’t a more fair comparison train K independent RGB models?   (2) When using multiple modalities, do you do any filtering? i.e., if the top box from depth and from edges is the same, do you use both as pseudo-groundtruth?
    - Appendix A – any idea why k is higher for RGB than depth and normals?
    - Table 3 – Given RGB helps improve for smaller classes, does it make sense to train a “Good-All” which uses RGB, depth and normals? Or to sample pseudo-groundtruth based on size (small boxes from RGB model, larger boxes from depth and normals)?  [No need to run this experiment, just curious about the authors’ thoughts]



**Summary Of The Paper:**

The authors present a self-training approach for detecting novel objects in an open-world setting using off-the-shelf monocular depth and normal estimators. The presented method improves over prior work on open-world settings on COCO and ADE20K.

**Summary Of The Review:**

Overall, this is a good paper that should be accepted. Given the nice idea and promising results, the work leaves me wanting more (results with instance segmentation, results on LVIS, further analysis, etc.), but the work passes the bar for acceptance as is.

---

> ### Author Response · Authors · 2022-11-17
> **Answer to Reviewer 3ZGc (1/2)**
>
> Thanks for your review and many constructive suggestions! We address your concerns as the following:
>
> **1. Segmentation.**
>
> Thanks for the suggestion! We agree that geometric cues can potentially lead to even more gains on segmentation. Unfortunately, due to time constraints, we cannot finish the experiments during the rebuttal period. We plan to include experiments regarding this in our final version.
>
> **2. Influence of the number of base classes.**
>
> We have included Figure 5c addressing this in our revised paper. We can see that GOOD is more effective at learning a generic sense of objectness and less prone to overfitting to the base classes — GOOD can achieve a similar AR_N @100 as OLN with only half the number of base classes, e.g., 39 vs. 80. As expected, we also observe decreasing performance gains as the number of base classes increases.  This is natural, as more object classes allow object detectors to learn a more generic sense of objectness, and can thus better detect novel objects. However, the annotation cost also grows along with the number of classes.
>
> **3. More benchmarks.**
>
> We have included LVIS as an additional cross-category benchmark and UVO as an additional cross-dataset benchmark. See Table 5 in our revised paper. Our results showed that GOOD has also surpassed previous state-of-the-art methods on these two benchmarks, e.g., GOOD improves AR_N from 27.4 (OLN) to 29.2 on the LVIS COCO to non-COCO benchmark, and improves AR_A from 49.2 (OLN) to 50.3 on the COCO to UVO benchmark.
>
> **4. More visualization of detection results on novel objects.**
>
> We have added more visualization examples of GOOD detection results in Figure 12. The test images contain objects that are seen neither in the GOOD training set (COCO) nor the Omnidata model training set. The presented examples include new technology devices, spaceships, dinosaurs, aliens, and sea scenes. We can see that GOOD can still make reasonable object bounding box predictions even though these objects have never appeared in the training set.
>
> **5. Original Figure 5.**
>
> We agree that a single example may lead to biased conclusions. We have removed it from the main paper.
>
> **6. Ensembling baselines and implementation in Figure 7b.**
>
> We agree that independently training K RGB models makes a fairer comparison. We have added this as an additional baseline (SelfTrain-RGB ens) to the figure and corrected the typo in the legend as well.
>
> For ensembling, we filter out the overlapping boxes. Specifically, if the IoU of two pseudo boxes is larger than 0.5, they are considered to be overlapping with each other, and the one with lower objectness score is filtered out. We have added these descriptions to the Appendix A.

---

> > ### Author Response · Authors · 2022-11-17
> > **Answer to Reviewer 3ZGc (2/2)**
> >
> > **7. The optimal choice of k for different modalities.**
> >
> > There is a trade-off when increasing k: while we can potentially include more novel objects, we also inject more noise into training. Since this trade-off is hard to analyze explicitly, we only have some hypotheses to understand the optimal choice of k.
> > - For RGB: since the model is more prone to overfitting  the training classes, the top predicted boxes could be very close to the training and not very valuable. So increasing the number might include more novel objects.
> > - For geometric cues: there is less training and test discrepancy, so the top 1 predictions might already be novel objects. Also, since geometric cues have estimation errors themselves, the pseudo boxes might also be noisier. So the benefit of increasing the number of pseudo boxes does not outweigh the potential loss.
> >
> > **8. Using RGB with geometric cues in Phase-I.**
> >
> > We have experimented with the suggested GOOD-All and put the results in Figure 6c. We find that adding pseudo boxes from RGB either leads to no performance gains or even worsens the performance, e.g., from GOOD-Both 34.0 AR_A to GOOD-All 33.3 AR_A on the COCO VOC to ADE20K benchmark. This is not surprising as SelfTrain-RGB underperforms good geometric cues across different benchmarks. From previous experiments and visualizations, we notice that models trained on RGB favor smaller detection boxes. We further validate this by plotting the histograms of the sizes of the pseudo boxes in Figure 6d. These smaller detection boxes can either be small objects or textures and parts of larger objects, which could potentially hurt the performance of the final detector to detect large objects. This is consistent with our observations in Table 6. In summary, for GOOD-All, the gains in AR small are usually too small to compensate for the losses in AR large, leading to inferior overall performance.
> >
> > Regarding sampling different sizes of boxes, we have also considered a similar method that aims to sample pseudo boxes of different sizes from RGB-based models (ScaleAug in Table 3) and found that it only provides marginal gains. Overall, we think that it is an interesting future extension to explore how to make better use of different types of pseudo boxes.
> >
> >
> > **9. Implementation details.**
> >
> > We have updated our description of the training details and used hyperparameters in Appendix A. We will release the code after the paper review phase.
> >
> > Thank you again for your comprehensive review (including the appendix) and your valuable suggestions!

---

> > ### Comment · Reviewer_3ZGc · 2022-11-24
> > **Thanks!**
> >
> > Thank you for your responses. I am satisfied with the responses, and maintain my rating that this paper should be accepted.

---

### Official Review · Reviewer_FTvx · 2022-10-25

**Confidence:** 4
**Clarity, Quality, Novelty And Reproducibility:** 1. This paper is well-written and eas…
**Correctness:** 3
**Technical Novelty And Significance:** 2
**Empirical Novelty And Significance:** 2
**Recommendation:** 6

**Strength And Weaknesses:**

[Strength]
1. This paper shows that depth and normals help overcome the overfitting problem of the only RGB-based proposal network, thus generating more unlabeled pseudo boxes which finally improve the detection performance.

[Weaknesses]
1. The novelty of this paper is limited. The network architecture and training losses used in this paper are the same as OLN. It looks that the only difference is that this paper uses additional geometric cues to train the proposal network, while OLN only uses RGB images. However, to my knowledge, geometric cues have been widely used in many 2D tasks to improve performance. Thus, this paper does not give new messages and insights.
2. In experiments, it is recommended to study how does the number of base classes influence the performance.

**Summary Of The Paper:**

This paper studied to improve the generalization of open-world class-agnostic object detection with geometric cues. In practice, depth and normal maps predicted from pretrained Omnidata model were used to train an object proposal network for pseudo-labeling unannotated novel objects. The resulting Geometry-guided Open-world Object Detector (GOOD) significantly improves detection recall for novel object categories even with only a few training classes.

**Summary Of The Review:**

I tend to reject this paper because of the limited novelty in this paper.

update: After discussion, I would like to raise my rating from 5 to 6 because of their great performance.

---

> ### Author Response · Authors · 2022-11-17
> **Answer to Reviewer FTvx**
>
> Thanks for your review! We address your concerns as the following:
>
> **1. The novelty of the paper.**
>
> As pointed out by the reviewer, our main contribution is the study of the utility of geometric cues for open-world class-agnostic object detection. As we discussed in Section 1 and 2 of the paper, all of the existing open-world object detectors are trained only on RGB images, resulting in overfitting to the training classes. By introducing geometric cues, we significantly mitigated the overfitting and improved the open-world object detection performance.  We also want to clarify that we are not aiming to extend the architecture or losses of the OLN. In fact, our approach is model-agnostic, and can in principle works with any object detectors. For example, we can also adopt a class-agnostic FCOS as the final detector in Phase-II. As shown in Table 7, when using the FCOS architecture, GOOD can also significantly improve the open-world object detection performance of the vanilla class-agnostic FCOS.
>
> As far as we know, our method is the first to demonstrate that estimated geometric cues can significantly improve detecting novel objects. We have also conducted extensive experiments to investigate the advantages of geometric cues. We hope our work can inspire future researchers to see the value of geometric cues and other mid-level representations in open-world tasks and out-of-distribution generalization tasks in general.
>
>
> **2. On the influence of the number of base classes**
>
> Thanks for the suggestion! We have added Figure 5c showing how the number of base COCO classes influences the performance on the ADE20K benchmark. We can see that GOOD is more effective at learning a generic sense of objectness and less prone to overfitting to the base classes — GOOD can achieve a similar AR_N @100 as OLN with only half the number of base classes, e.g., 39 vs. 80. As expected, we also observe decreasing performance gains as the number of base classes increases. This is natural, as more object classes allow object detectors to learn a more generic sense of objectness, and can thus better detect novel objects. However, the annotation cost also grows along with the number of classes.
> We have also added more ablation studies on the ensembling choices in Appendix C.
>
> Thank you again for your review and your valuable suggestions!

---

> ### Author Response · Authors · 2022-11-30
> **More questions?**
>
> Dear Reviewer FTvx,
>
> Thank you again for your review. We hope our rebuttal can satisfactorily address all your questions and concerns regarding the novelty of the paper and the experiments on the influence of the number of base classes. We wonder if you still have any concerns we could address.
>
> Thank you for your time.

---

> > ### Comment · Reviewer_FTvx · 2022-12-03
> > **Thanks for your rebuttal**
> >
> > After reading rebuttals and the comments by other reviewers, I still have some concerns about the novelty of using depth (and normal) cues for open-world object detection, which has been testified by many previous studies. For example, the paper "Learning Rich Features from RGB-D Images for Object Detection and Segmentation" in ECCV 2014 has presented a detection system that takes an RGB and depth image pair for object detection. Therefore, in my opinion, simply introducing the depth (and normal) cues for (open-world) object detection is not that new.

---

> > > ### Author Response · Authors · 2022-12-03
> > > **Answers to follow-up concerns**
> > >
> > > Dear reviewer FTvx,
> > >
> > > Thanks for your new response. Regarding your concerns about novelty, we want to discuss two points:
> > >
> > > 1. **Open-world object detection is a different task from standard object detection as it deals with out-of-distribution data.** Many works, e.g., [1-3] have reported that observations on standard models can change under OoD. For the open-world object detection problem, our work is the first to incorporate inferred geometric cues and achieve significant performance gains. Our key insight is that while the inferred depth and normals as additional input modalities cannot directly improve open-world object detection, **they can still discover novel objects that RGB-based methods cannot discover, which can be used to boost performance through pseudo-labeling**. This is specific to the open-world problem where the model has to deal with unseen objects, and sets us distinctively different from previous RGB-D based methods that use depth directly.
> > >
> > > 2. **Incorporating inferred geometric cues to improve open-world object detection is not trivial.** Current RGB-D-based methods use groundtruth depth to train the object detector and only focus on in-distribution test performance. Our method differs from them in two aspects: (a) we only use inferred depth and normals, which contain noise and are more difficult to incorporate; (b) we tackle the open-world (out-of-distribution) problem, which is underexplored by current methods. As shown in our experiments (Fig 5, Fig 6), it is not trivial to incorporate geometric cues with RGB inputs. Our proposed pseudo-labeling method mitigates the potential harm from estimation noise of geometric cues and also exploits unannotated objects in the training data, making it a simple and effective way to significantly surpass previous methods. Therefore, we think our method is innovative and can inspire future works to further explore inferred geometric cues for out-of-distribution generalization in computer vision.
> > >
> > > **References**
> > >
> > > [1] Joseph, K. J., et al. "Towards open world object detection." Proceedings of the IEEE/CVF Conference on Computer Vision and Pattern Recognition. 2021.
> > >
> > > [2] Zhao, Bingchen, et al. "OOD-CV: A Benchmark for Robustness to Out-of-Distribution Shifts of Individual Nuisances in Natural Images." European Conference on Computer Vision. Springer, Cham, 2022.
> > >
> > > [3] Hendrycks, Dan, et al. “Benchmarking Neural Network Robustness to Common Corruptions and Perturbations.” International Conference on Learning Representations. 2019.

---

### Official Review · Reviewer_SvRu · 2022-10-28

**Confidence:** 5
**Correctness:** 3
**Technical Novelty And Significance:** 3
**Empirical Novelty And Significance:** 2
**Recommendation:** 6

**Clarity, Quality, Novelty And Reproducibility:**

Most of the paper is clearly written. However, as mentioned above, the authors should add more description/details about some of their decisions regarding the model design. The quality of the paper is good overall but can be significantly improved by incorporating the suggestions above. The proposed approach is novel and can benefit a large portion of the computer vision community. The idea seems simple enough to be reproducible, but I will appreciate it if the authors release their code.

**Strength And Weaknesses:**

The idea of explicitly using geometric cues for object proposal generation is novel and interesting. The authors have proposed a two phase training pipeline which first uses geometric cues to generate proposals for unlabeled objects. These pseudo-labels and the original ground-truths are used to fine-tune the proposal network in the second phase. The authors show that the proposed approach can achieve improved performance.

The main weaknesses of the paper are lack of details about the decisions, and an absence of discussion on how to extend the proposed method to more recent proposal-free object detection methods. In particular, the authors should address the following in a revision/rebuttal:

1. How exactly is the proposal network trained in Phase 1? Is the same network trained for both depth and normal? How exactly? Are the depth and normal maps stacked? Or are they just used as separate/independent images?

2. Connected with the first point above - Why not use RGB images as well in the first step, along with depth and normal? Does that lead to worse performance? Why/Why not? Further, what other geometric cues can be used in addition to depth and normal?

3. I want the authors to consider and discuss, how can this proposed approach be extended to proposal-free object detectors? What is the utility of proposal-based object detectors if DETR and other similar object detectors give better performance? Further, I want the authors to consider if the presented approach can help proposal-based object detectors achieve a higher performance than more recent proposal-free object detectors. For example, can a Mask-RCNN containing an RPN trained with the proposed approach get better performance than a DETR model?

4. The authors should demonstrate the effectiveness of the proposed approach on larger and more challenging datasets like LVIS. This will help in making their claims stronger.



After discussion with the authors and consulting with the AC and other reviewers, I am increasing my rating.

**Summary Of The Paper:**

This paper proposes an interesting and novel idea of using geometric cues like depth and normal maps to obtain object proposals for object detection. The authors propose a two phase approach for improving object proposals for unseen objects. In the first stage they train an object proposal generation method which uses geometric cues like depth and normal maps to provide pseudo-labeling. The generated pseudo-labels are added to the training set and the final proposal method is trained using the original boxes and these generated boxes.

**Summary Of The Review:**

I am currently recommending a "weak-reject" rating based on the weaknesses mentioned above. However, I would be happy to consider upgrading it to an "accept" if the authors are able to satisfactorily address the points mentioned above.

---

> ### Author Response · Authors · 2022-11-17
> **Answer to Reviewer SvRu (1/2)**
>
> Thanks for the review and the questions! We address your concerns as the following:
>
> **1. Training details.**
>
> We trained the proposal network with the training loss as given in eq. (2) and used the SGD optimizer with an initial learning rate of 0.01 and batch size of 16 for 8 epochs. We trained separate models (with the same architecture) on depth and normal maps, and generated two separate sets of pseudo boxes. For GOOD-Both, we merge these pseudo boxes by filtering out the overlapping boxes. Specifically, if the IoU of two pseudo boxes is larger than 0.5, they are seen as overlapping with each other, and the one with a lower objectness score will be filtered out. We have added these details in Appendix A.
>
> Another choice of using depth and normals together is to stack the two maps together and train a single model. We have also experimented with this option. We found that this choice can also achieve similar (but slightly worse) results as our chosen training strategy. See the newly added ablation experiment (Figure 6a) in the Appendix.
>
> **2a. Using RGB in Phase-I.**
>
> There are two possible ways to use RGB in Phase-I with geometric cues: (1) Stack RGB inputs with the two geometric cues together and train a single object proposal network on these stacked inputs in Phase-I; (2) Train three separate object proposal networks and extract pseudo boxes separately, then merge them into a single pseudo box pool for Phase-II training. From our experiments (Figure 6), neither shows stronger performance than only using geometric cues in Phase-I.
>
> If we directly stack RGB with depth and normal maps to train the object proposal network in Phase-I, the model can still choose to take the shortcut and rely on the appearance cues from the RGB inputs. This would be against our primary intention of exploring geometric cues. From our experiments (Figure 6b), stacking RGB with geometric cues indeed leads to worse final performance than stacking geometric cues only for training the proposal network.
>
> An alternative way to use RGB in Phase-I is to separately train a proposal network on RGB in Phase I and then merge the generated pseudo boxes with those from models trained on the geometric cues for Phase II training. We name this method “GOOD-All”.
> From the experiments (Figure 6c), we find that adding pseudo boxes from RGB either leads to no performance gains or even worsens the performance, e.g., from GOOD-Both 34.0 AR_A to GOOD-All 33.3 AR_A on the COCO VOC to ADE20K benchmark. This is not surprising as SelfTrain-RGB underperforms GOOD with geometric cues across different benchmarks.  From previous visualizations (Figure 10, 11), we notice that models trained on RGB favor smaller detection boxes. We further validate this by plotting the histograms of the sizes of the pseudo boxes in Figure 6d. These smaller detection boxes can either be small objects or textures and parts of larger objects, which could potentially hurt the performance of the final detector to detect large objects. This is consistent with our observations in Table 6. In summary, for GOOD-All, the gains in AR small are usually too small to compensate for the losses in AR large, leading to inferior overall performance.
>
> **2b. Using other geometric cues.**
>
> Further, in principle, our method can incorporate other geometric cues as well, such as reshading, curvature, and occlusion edges. All these show different aspects of scene geometry and can potentially help discover new objects. In our paper, we focused on depth and normals, thanks to the strong pretrained Omnidata depth and normal estimation models.
> Relatedly, we experimented with 2D edges in our paper (Table 2 and 8) and found that depth and normals demonstrate much stronger performance than it. This could be due to the fact that the edge map (extracted using HED) does not have as high quality as the depth and normal maps extracted from Omnidata models.
>
> In the future, it could be interesting to explore other geometric cues and ways of incorporating them further to enhance the performance of the open-world object detectors.

---

> > ### Author Response · Authors · 2022-11-17
> > **Answer to Reviewer SvRu (2/2)**
> >
> > **3. On the architecture choice of the object detector.**
> >
> > One advantage of our pseudo-labeling approach is that it is model agnostic, thus it is compatible with both proposal-based and proposal-free object detectors.
> >
> > For Phase-I, the depth or normal maps can be the input for any architecture type of object detectors and trained in a class-agnostic manner. The pseudo box generation and merging with gt ones are independent of the model choices in Phase-I. Once the new annotation set is constructed, it can be used for training any object detector in Phase-II. The models in Phase-I and Phase-II can also adopt different architectures. The proposal network in Phase-I is only for extracting pseudo boxes and is used neither during Phase-II training nor at inference time. Only the object detector in Phase-II is used as the final open-world object detector.
> >
> > Therefore, while we followed the practice in the related work OLN and GGN to demonstrate the effectiveness of our approach using a two-stage proposal-based object detector (FRCNN), our method is not limited to this architecture.
> >
> >
> > To further demonstrate this, we experimented with a more recent proposal-free object detector in Phase-II, namely FCOS [1]. From Table 7, we can see that our approach can also enhance FCOS, e.g., from AR_N 29.3 to 36.3 on COCO VOC to non-VOC benchmark. Further, we find that our approach achieves the best performance when combined with the proposal-based OLN (a modified FRCNN architecture) rather than FCOS, e.g., GOOD+OLN achieves 39.3 AR_N on COCO VOC to non-VOC benchmark, while GOOD+FCOS achieves 36.3.
> >
> >
> >
> > **4. More benchmarks like LVIS.**
> >
> >  We have included results on LVIS and UVO in Table 5 of our revised paper. Our results showed that GOOD has also surpassed previous state-of-the-art methods on these two benchmarks, e.g., GOOD-Normal improves AR_N from 27.4 (OLN) to 29.2 on the LVIS COCO to non-COCO benchmark, and GOOD-Depth improves AR_A from 49.2 (OLN) to 50.3 on the COCO to UVO benchmark.
> >
> > Thank you again for your comprehensive review and your valuable suggestions!
> >
> >
> > **References**
> >
> > [1] Tian, Zhi, et al. "FCOS: Fully convolutional one-stage object detection." Proceedings of the IEEE/CVF international conference on computer vision. 2019.

---

> > > ### Comment · Reviewer_SvRu · 2022-11-20
> > > **Response to Authors' Rebuttal**
> > >
> > > I thank the authors for their responses to my initial review. Though the authors have tried to address my questions, I still have a few concerns about the paper and the response.
> > >
> > > 1. The authors say, "If we directly stack RGB with depth and normal maps to train the object proposal network in Phase-I, the model can still choose to take the shortcut and rely on the appearance cues from the RGB inputs". I don't understand why is this. Why would RGB be a "short-cut" and how would a model select it?
> > >
> > > 2. I don't understand why the performance with RGB is lower in both cases compared to just using the geometric cues. In both cases, the geometric information is still present.
> > >
> > > 3. My biggest concern is still the proposed approach can be applied to the latest Transformer-based object detectors. If yes, how? And how much performance improvement can be expected?

---

> > > > ### Author Response · Authors · 2022-11-23
> > > > **Answers to follow-up questions from Reviewer SvRu (1/2)**
> > > >
> > > > Thanks for taking the time to continue discussing with us; we appreciate it! We would like to address your further concerns as the following:
> > > >
> > > > **1. Why would RGB be a shortcut**
> > > >
> > > > As defined in [1], shortcuts are decision rules that perform well on i.i.d. test data (**closed-world assumption**) but fail on o.o.d. tests (**open-world assumption**), revealing a mismatch between intended and learned solution. For example, a test image of “a cat but with elephant textures” can be classified as “an elephant” for a model that learns texture shortcuts from the training set [2].
> > > >
> > > > In our case, when we stack RGB with geometric cues to train the proposal network, **the model will tend to make more use of RGB to optimize the target localization loss**. This is because RGB is a much stronger input signal than geometric cues in the closed-world setup — AR100_base is 58.3 for RGB inputs alone and 44.9 when stacking depth and normals on COCO VOC classes. In the extreme case, the model can even completely ignore geometric cues. This reliance on RGB inputs prevents the model from making the best use of geometric cues to discover novel objects in Phase-I, which is crucial for open-world object detection. Therefore, we say that using RGB inputs is a shortcut for our case.
> > > >
> > > > Regarding “how would a model select the shortcut”, it is generally an open question. Intuitively, this can be understood using the “Principle of Least Effort” [3], which means the model will find the easiest ways to optimize its target loss. In our case, using RGB to optimize the training loss is considerably easier than using geometric cues. Therefore, the model tends to take the shortcut and rely on the appearance cues of RGB inputs, such as textures of objects.
> > > >
> > > > **2. On incorporating RGB in Phase-I.**
> > > >
> > > > (a)  If we stack RGB with geometric cues and train a single model, as explained in the previous point, due to the short-cut learning, the top predictions from this model may heavily rely on RGB signals, which means the geometric information may not be left much in Phase-II.
> > > >
> > > > (b) If we merge the pseudo labels of the three models, then the generalization performance of the final object detector depends on the overall quality of these pseudo boxes. It is known in semi-supervised learning that more pseudo boxes don’t necessarily improve the final performance [4][5]. Noisy (low-quality) pseudo boxes can actually worsen the performance.
> > > >
> > > > Therefore, **if the quality of pseudo boxes from RGB is low, it can lead to performance deterioration, even if the geometric information is unchanged compared to GOOD-Both**. We can imagine the extreme case: if the pseudo box from RGB is randomly generated, then it will surely provide low-quality supervision, making the model perform worse. So, when pseudo boxes from RGB introduce more noise than useful information, it can lead to performance deterioration.
> > > >
> > > > In our initial reply and Appendix C.2, we found that adding pseudo boxes from RGB will lead to performance gains on AR_small, but performance losses on AR_large. In Fig. 6b and Fig.10-11,  we observed that pseudo boxes from RGB contain more smaller detection boxes which can correspond to parts of larger objects, potentially hurting the detection of large objects. Overall, we found that the performance gain on AR_small cannot compensate for the losses on AR_large, leading to overall worse performance.

---

> > > > > ### Author Response · Authors · 2022-11-23
> > > > > **Answers to follow-up questions from Reviewer SvRu (2/2)**
> > > > >
> > > > > **3. Can GOOD be applied to transformer-based object detectors?**
> > > > >
> > > > > We conducted an additional experiment using the DN-DETR [6]. We used the 12-epoch setup in their official codebase to train a class-agnostic DN-DETR and a GOOD+DN-DETR on COCO VOC classes. Like FCOS in Appendix C.3, when applying GOOD, we also use DN-DETR in Phase-II as the final object detector. **Our results show that GOOD can also significantly improve DN-DETR on COCO VOC to non-VOC benchmark (from AR100_novel 13.7 to 28).** Since transformer-based methods typically need more training data, we hypothesize that their advantage when there are fewer annotations is not large compared to other architectures.
> > > > >
> > > > > We noticed that in the original review, the reviewer wrote that “These pseudo-labels and the original ground-truths are used to fine-tune the proposal network in the second phase.”  in the first paragraph of the “Strength and Weaknesses” section. We want to clarify that the “proposal network” is **NOT** finetuned, and is also **NOT** the RPN of Faster RCNN. Instead, a new object detector is trained in Phase-II. In fact, the proposal network in Phase-I can use any model architecture such as RPN, Faster RCNN, FCOS, and DETR. The proposal network is only used to provide the pseudo boxes which are then used to train the final object detector, which can also adopt any architecture of object detectors.
> > > > >
> > > > > **References**
> > > > >
> > > > > [1]Geirhos, Robert, et al. "Shortcut learning in deep neural networks." Nature Machine Intelligence 2.11 (2020): 665-673.
> > > > >
> > > > > [2] Geirhos, Robert, et al. Imagenet-trained CNNs are biased towards texture; increasing shape bias improves accuracy and robustness. In International Conference on Learning Representations, 2019.
> > > > >
> > > > > [3] Zipf, George Kingsley. Human behavior and the principle of least effort: An introduction to human ecology. Ravenio Books, 2016.
> > > > >
> > > > > [4] Li, Hengduo, et al. "Rethinking pseudo labels for semi-supervised object detection." Proceedings of the AAAI Conference on Artificial Intelligence. Vol. 36. No. 2. 2022.
> > > > >
> > > > > [5] Li, Gang, et al. "Pseco: Pseudo labeling and consistency training for semi-supervised object detection." European Conference on Computer Vision. Springer, Cham, 2022.
> > > > >
> > > > > [6] Li, Feng, et al. "Dn-detr: Accelerate detr training by introducing query denoising." Proceedings of the IEEE/CVF Conference on Computer Vision and Pattern Recognition. 2022.

---

> > > > ### Author Response · Authors · 2022-11-30
> > > > **More questions?**
> > > >
> > > > Dear Reviewer SvRu,
> > > >
> > > > Thank you again for your review. We hope our rebuttal can satisfactorily address all your questions and concerns regarding the proposal network, usage of RGB inputs, and results on more datasets. We wonder if you still have any concerns we could address.
> > > >
> > > > Thank you for your time!

---

### Public Comment · ~Yuchen_Wu3 · 2022-11-06
**Questions About Open World Detection on Depth Image**

Hi, I'm also working on the direction of open-world proposal and trying to reproduce the result in your paper. I use the V2 model of OmniData and inference on the padded COCO image according to the paper. Then I train an OLN model on the depth images by directly loading the depth images as 3-channel arrays (duplicating the depth channel). The model is trained for 8 epochs using the same schedule as the OLN code. But I only get AR@100=23.4, far from the result you provide in Fig6a, which is 27.7. Could you please provide more details on training the proposal network on depth so that I can check if there's anything that I missed (e.g. the data aug)? Thank you very much.

---

> ### Author Response · Authors · 2022-11-06
> **Please check your implementation of extracting the depth images**
>
> Hi Yuchen, thanks for your attention on our work! Regarding your implementation, we think you can check the following items:
>
> (1) What’s the size of the depth maps you are extracting? Did you keep the original sizes of the COCO images? Note in the demo file of the Omnidata tools [1], the default image size is 374, which is notably smaller than the COCO image sizes and will lead to inferior training performance. Besides using the original size of COCO images (with padding), you can also try resizing it to 576 *576 or 640 * 640. In our experiments, we find they all work similarly well. Also, please check whether you have sliced the extracted outputs to the original image size, i.e., reverse the padding. For this you can simply use `output= output[:, :height, :width]`. Further, please also make sure you have implemented the data loader correctly. Specifically, you need to match the input processing of the Omnidata model (instead of OLN's). These include steps like normalization and using RGB channels instead of BGR. For this, you can check their training code [1].
>
> (2) How do you save the depth images? Specifically, please refer to the aforementioned demo file for reference. Note there is a bicubic interpolation (if you are not using the original shape) and a clamp function for truncation.  If you have used the code from the demo file, you shouldn’t need to duplicate the depth maps since they are saved 3-channel images already by mapping to colormaps.
>
> To train the proposal network, we have exactly the same hyperparameters as OLN, which you are already using.
>
> Please let us know if you have further trouble reproducing our results. We will provide further support after the rebuttal and release the code as well.
>
> [1] Code repository of Omnidata tools (Pytorch): https://github.com/EPFL-VILAB/omnidata/tree/main/omnidata_tools/torch

---

> > ### Public Comment · ~Yuchen_Wu3 · 2022-11-06
> > **The Storage of Depth Image**
> >
> > Thank you for your quick response! And I hope I didn't bother you during the rebuttal process :) . Here is my modification to the omnidata
> > demo.py script for the above-mentioned experiments. Note that instead of saving it using the mapped color maps, I saved the depth as a single-channel PNG file. According to your reply, I will try again using the mapped color image.
> > ```python
> > width, height = img.size
> > width_pad = math.ceil(width / 32) * 32
> > height_pad = math.ceil(height / 32) * 32
> > img_tensor_pad = F.pad(img_tensor, (0, width_pad - width, 0, height_pad - height), mode='reflect')
> > img_tensor_pad = img_tensor_pad[:3].unsqueeze(0).to(device)
> >
> > output = model(img_tensor_pad).clamp(min=0, max=1)
> > output = output[:, :height, :width]
> > output = output*255
> > array = output.detach().cpu().squeeze().numpy().astype(np.uint8)
> > im = Image.fromarray(array)
> > im.save(save_path, "PNG")
> > ```

---

> > > ### Public Comment · ~Yuchen_Wu3 · 2022-11-06
> > > **Update on Reimplementation**
> > >
> > > Thank you very much. I can now get similar results using the mapped 3 channel images. Maybe it was because my original implementation of 8 bit single-channel depth image loses too much information.

---

### Author Response · Authors · 2022-12-05
**General response -- thanks to all reviewers for their valuable feedback**

We’d like to thank all reviewers again for their valuable feedback. In the following, we will use R1-R4 to refer to the reviewers in the order of their reviews.

We are pleased to see that reviewers found that our proposed method is **novel** (R1, R3, R4) and **effective** (R1, R2, R3, R4), and **demonstrates the value of geometric cues** (R3, R4) which are underexplored for the open-world problems. We are equally glad that the reviewers found the paper **well-written** and **easy to understand** (R1, R2, R3, R4).

We summarized our responses to the major concerns of the reviewers as the following:

1. We have added **results on more benchmarks** (R1, R3, R4), namely LVIS COCO to non-COCO and COCO to UVO, to Table 5 of our revised paper. Our results showed that GOOD has also surpassed previous state-of-the-art methods on these two benchmarks.
2. We clarify that our method is **agnostic to architecture choice** and demonstrate that it can still **make significant improvements using other network architectures** such as FCOS and DN-DETR (R1).
3. We have also provided **a comprehensive analysis of different ways to use RGB in Phase-I** (R1 & R3) and showed that GOOD-Both, i.e., merging pseudo boxes from proposal networks trained depth and normals, provided the most robust and superior performance across different benchmarks.
4. Further, we have also provided **experiments on the influence of the number of base classes** (R2 & R4). The results showed that our method can achieve similar performance as RGB-based methods with less than half of the training classes, thus saving a lot of potential annotation costs.
5. Finally, we have added **more implementation details** for reproducibility (R1 & R3).

We thank R3 for acknowledging our response that addressed the concerns and keeping the rating of 8 to recommend for acceptance. We appreciate the further post-rebuttal discussion with R1 and R2. We invite **R1** to check the new results on DN-DETR which showed that GOOD can also significantly improve DN-DETR,   and our detailed follow-up explanation regarding the use of RGB input. We provide further clarification regarding the paper’s novelty to **R2**. We believe we have also addressed all concerns from **R4**, i.e., LVIS and UVO benchmarks and the scenario when the training set is (nearly) exhaustively annotated. The experiment results are positive. Thus, we invite **R1, R2, and R4** to check our responses and consider improving the scores if finding them convincing.

---

### Decision · Program_Chairs · 2023-01-20

**Decision:**

Accept: poster

**Justification For Why Not Higher Score:**

Reviewers recommended poster; scores seem indicative of poster also.

**Justification For Why Not Lower Score:**

Following the discussion, all reviewers were supportive of (or at least not opposed to) acceptance.

**Metareview: Summary, Strengths And Weaknesses:**

This paper proposes to use geometric (depth, normals) information to cope with unlabeled objects in open-world object detection. Reviewers appreciate the insights from this paper, namely that depth enables generalization to new objects, as well as the strong results, including the additional results with DETR, and on LVIS. They are concerned about technical novelty and limited gains with more classes, but in the virtual meeting, found both to not be critical.

**Note From Pc:**

if the above contains the word "oral" or "spotlight" please see: "oral" presentation means -> notable-top-5% and "spotlight" means -> notable-top-25%. As stated in our emails, we are disassociating presentation type from AC recommendations

**Summary Of Ac-Reviewer Meeting:**

Two reviewers indicated they'd like to increase their scores. One question that arose is whether Omnidata can be used to predict depth for images at test time, rendering a setting similar to RGB-D detection which is known.